# A periplasmic cinched protein is required for siderophore secretion and virulence of *Mycobacterium tuberculosis*

Lei Zhang [1], James E. Kent [2], Meredith Whitaker[3], David C. Young[4], Dominik Herrmann [1], Alexander E. Aleshin[2], Ying-Hui Ko[5], Gino Cingolani [5], Jamil S. Saad [1], D. Branch Moody [4], Francesca M. Marassi[2], Sabine Ehrt [3] & Michael Niederweis [1]✉

Iron is essential for growth of *Mycobacterium tuberculosis*, the causative agent of tuberculosis. To acquire iron from the host, *M. tuberculosis* uses the siderophores called mycobactins and carboxymycobactins. Here, we show that the *rv0455c* gene is essential for *M. tuberculosis* to grow in low-iron medium and that secretion of both mycobactins and carboxymycobactins is drastically reduced in the *rv0455c* deletion mutant. Both water-soluble and membrane-anchored Rv0455c are functional in siderophore secretion, supporting an intracellular role. Lack of Rv0455c results in siderophore toxicity, a phenotype observed for other siderophore secretion mutants, and severely impairs replication of *M. tuberculosis* in mice, demonstrating the importance of Rv0455c and siderophore secretion during disease. The crystal structure of a Rv0455c homolog reveals a novel protein fold consisting of a helical bundle with a 'cinch' formed by an essential intramolecular disulfide bond. These findings advance our understanding of the distinct *M. tuberculosis* siderophore secretion system.

[1] Department of Microbiology, University of Alabama at Birmingham, Birmingham, AL 35294, USA. [2] Cancer Center, Sanford Burnham Prebys Medical Discovery Institute, La Jolla, CA 92037, USA. [3] Department of Microbiology and Immunology, Weill Cornell Medical College, New York, NY 10021, USA. [4] Division of Rheumatology, Inflammation and Immunity, Brigham and Women's Hospital, Harvard Medical School, Boston, MA 02115, USA. [5] Department of Biochemistry and Molecular Biology, Thomas Jefferson University, Philadelphia, PA 19107, USA. ✉email: mnieder@uab.edu

Iron is an essential element for almost all living organisms as it is utilized in a wide range of metabolic processes, including oxygen transport, DNA synthesis, and electron transport[1]. However, free iron is not easily accessible for microbes in the presence of oxygen and at neutral pH[2]. In vertebrates, almost all iron is tightly bound to host proteins such as ferritin, transferrin, lactoferrin, hemoglobin, cytochromes, or iron-sulfur cluster proteins[3], reducing the iron availability and limiting the replication of pathogenic bacteria, especially of chronic pathogens with long in vivo life cycles[4]. To overcome iron limitation, *Mycobacterium tuberculosis*, the causative agent of tuberculosis, acquires iron from heme[5] and solubilizes Fe(III) using hydroxyphenyl-oxazolone siderophores, namely the hydrophobic mycobactins and the water-soluble carboxymycobactins, which share the same high-affinity iron-binding core[6]. While mycobactins are mainly membrane-associated and carboxymycobactins are secreted, both siderophores are detected in the culture filtrate of *M. tuberculosis*[7,8] demonstrating that not only carboxymycobactins but also mycobactins are secreted. This is consistent with the observation that lipid vesicles secreted by *M. tuberculosis* contain mycobactin[9]. Importantly, addition of mycobactin or carboxymycobactin to the culture medium rescues the growth of an *M. tuberculosis* mycobactin synthesis mutant in low-iron medium demonstrating that both classes of siderophores are functionally identical in iron acquisition[10]. These siderophores are essential for growth of *M. tuberculosis* in macrophages[11]. During their synthesis, MBTs and carboxymycobactins are maintained in an iron-free state in their deoxy forms until the iron-binding sites are activated by enzyme-mediated oxidation[12]. The active siderophores are then exported across the inner membrane by the efflux pumps MmpL4 and MmpL5 and their associated MmpS4 and MmpS5 proteins[7,10]. Lack of both *mmpS4* and *mmpS5* genes abolishes siderophore secretion and results in strong attenuation of virulence of *M. tuberculosis* in mice[7].

However, it is unclear how mycobactins and carboxymycobactins are transported through the periplasm and across the outer membrane to function as siderophores. Proteins mediating these steps required for siderophore secretion are unknown and cannot be identified by sequence, since *M. tuberculosis* does not have homologs to TolC, which provides a periplasmic tunnel by docking to various efflux pumps via periplasmic adapter proteins and an outer membrane channel for secretion of the siderophore enterobactin by *E. coli*[13,14].

In this study, we show that the mycobacterial core protein Rv0455c, a 14-kDa exported protein of previously unknown function, is required for siderophore secretion by *M. tuberculosis*. Similar quantities of Rv0455c protein are localized in the periplasm of *M. tuberculosis* and are secreted into the culture medium, but Rv0455c exerts its function in siderophore secretion in the periplasm. The crystal structure of the Rv0455c homolog MSMEG_3494 from *M. smegmatis* reveals a novel structure consisting of a helical bundle with a cinch topology established by a disulfide bond formed by two essential cysteines. *M. tuberculosis* lacking *rv0455c* is attenuated in mice providing evidence that single proteins required for siderophore secretion exist and validating siderophore secretion as a target for tuberculosis drugs with a new mechanism of action. This study represents a major step forward in understanding the novel *M. tuberculosis* siderophore secretion system.

## Results

### Deletion of *rv0455c* gene renders *M. tuberculosis* hypersensitive to exogenous mycobactin or carboxymycobactin

In a previous Tn-seq study, we found that *M. tuberculosis* does not tolerate transposon insertions in the *rv0455c* gene in medium containing mycobactin even in the presence of hemin, while *rv0455c* is dispensable for growth of *M. tuberculosis* with hemin or hemoglobin as sole iron sources[10]. This phenotype is similar to that observed for the siderophore secretion-deficient *mmpS4/mmpS5* deletion mutant of *M. tuberculosis*[15] and was termed siderophore poisoning[10]. To investigate whether Rv0455c is involved in siderophore secretion by *M. tuberculosis*, the *rv0455c* gene was deleted in both the avirulent *M. tuberculosis* strain mc²6230 (H37Rv ΔRD1 ΔpanCD) and the virulent *M. tuberculosis* H37Rv strain by homologous recombination (Fig. S1a–c). The *M. tuberculosis* Δ*rv0455c* mutant ML2203 (Table S1) derived from *M. tuberculosis* mc²6230 hardly grew in low-iron 7H9 medium (<0.1 μM Fe³⁺). In contrast, both the parent strain or *M. tuberculosis* ML2203 complemented with an integrative vector expressing *rv0455c* showed logarithmic growth and stationary phase survival (Fig. 1a). The growth defect was similar to that observed for the *M. tuberculosis* strains lacking the *mmpS4/mmpS5* genes (*M. tuberculosis* ML859) or the mycobactin synthesis gene *mbtD* (*M. tuberculosis* ML1600) (Table S1). Further, the Δ*rv0455c* mutant, as well as the siderophore secretion-deficient mutant Δ*mmpS4*/Δ*mmpS5*, exhibited significantly reduced growth in the self-made 7H9 medium with Fe³⁺ concentrations ranging from 1 μM to 100 μM, compared to *M. tuberculosis* mc²6230 (Fig. S1d), indicating that the ability of both mutants to utilize iron salts was compromised. The growth defect of all mutants in low-iron medium was partially rescued by supplementing 10 μM hemin as an alternative iron source (Fig. 1b). However, addition of 100 nM mycobactin or 1 μM carboxymycobactin to the medium containing 10 μM hemin completely inhibited the growth of the Δ*rv0455c* mutant, in contrast to that of the mycobactin synthase D (*mbtD*) mutant, which lacks mycobactin and carboxymycobactin, but similar to that of the Δ*mmpS4*/Δ*mmpS5* mutant (Fig. 1c, d). The rescue of *mbtD* mutant with mycobactin or hemin is consistent with reversing an iron-deficient state in *M. tuberculosis*, but arrested growth of the Δ*rv0455c* mutant with added siderophores in the presence of hemin is the hallmark of siderophore poisoning as previously observed for the siderophore secretion-defective mutants lacking either *mmpS4* and *mmpS5*[10,15] or *mmpL4* and *mmpL5*[10]. In addition, we showed that the siderophore poisoning phenotype is independent of the ESX-1 system, as shown for the Δ*rv0455c* mutant directly derived from the virulent *M. tuberculosis* H37Rv parent strain (ML2700; Table S1) in the growth assays with mycobactin or carboxymycobactin (Fig. S1e–g).

To determine the minimal inhibitory concentration of exogenous siderophores for the Δ*rv0455c* mutant, we assessed its viability in the presence of 20 μM hemin by the microplate Alamar Blue assay (MABA). Dose-dependent growth inhibition by mycobactin and carboxymycobactin was observed for the Δ*rv0455c* mutant. Mycobactin was ~20-fold more toxic (MIC₉₀ ~ 50 nM) for the Δ*rv0455c* mutant compared to carboxymycobactin (MIC₉₀ ~ 1 μM). Similar differences were observed previously for the Δ*mmpS4/S5* mutant[15] (MIC₉₀ ~ 25 nM for mycobactin and 500 nM for carboxymycobactin) (Fig. 1e, f). Taken together, these experiments showed that the *M. tuberculosis* Δ*rv0455c* mutant is hypersensitive to exogenous mycobactins and carboxymycobactins, indicating dysfunction or mislocalization of siderophores in the absence of Rv0455c.

### Rv0455c homologs are functionally conserved in mycobacteria

Rv0455c is annotated as a protein of unknown function containing the DUF5078 domain (Pfam database[16]). Recognizable Rv0455c homologs are found only in *Mycobacterium* and other closely related genera such as *Segniliparus* with an outer

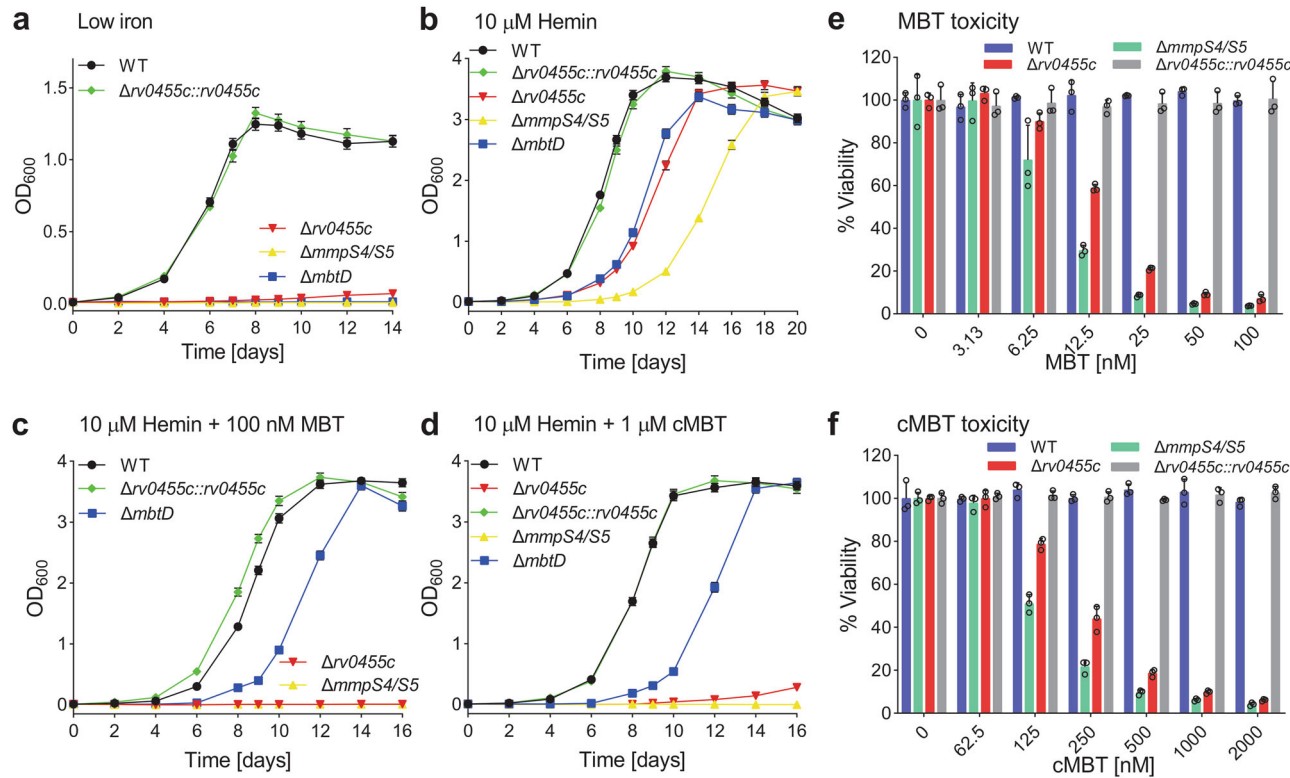

**Fig. 1 The *M. tuberculosis* Δ*rv0455c* mutant is hypersensitive to exogenous siderophores. a–d** The indicated *M. tuberculosis* strains were grown in low-iron 7H9 medium for 5 days to deplete intracellular iron before the growth assays. The initial $OD_{600}$ of all cultures was 0.01. Growth curves of *M. tuberculosis* mc$^2$6230 (wt), Δ*rv0455c* (*M. tuberculosis* ML2203), Δ*rv0455c::rv0455c* (*M. tuberculosis* ML2205), Δ*mmpS4/S5* (*M. tuberculosis* ML859) and Δ*mbtD* (*M. tuberculosis* ML1600) in (**a**) self-made low-iron (<0.1 μM $Fe^{3+}$) 7H9 medium supplemented with (**b**) 10 μM hemin, (**c**) 10 μM hemin plus 100 nM Fe-mycobactin (Fe-MBT), and (**d**) 10 μM hemin plus 1 μM Fe-carboxymycobactin (Fe-cMBT), respectively. Error bars represent standard deviations from the mean results of biological triplicates (*n* = 3). **e**, **f** The viability of *M. tuberculosis* wt, Δ*mmpS4/S5*, Δ*rv0455c*, or Δ*rv0455c::rv0455c* was measured using the Microplate alamarBlue assay. Cells treated with increasing concentrations of (**e**) Fe-mycobactin (Fe-MBT) or (**f**) Fe-carboxymycobactin (Fe-cMBT). Cells were grown in 7H9 medium supplemented with 20 μM hemin. Error bars represent standard deviations from the mean values of biological triplicates (*n* = 3). Source data are provided in the Source Data file.

membrane composed of long-chain mycolic acids (C60-C100)[17], whereas the outer membranes of Gram-negative bacteria consist mainly of lipopolysaccharides and phospholipids. It is noteworthy that *rv0455c* and its homologous genes are always neighboring the *mmpS4-mmpL4* or *mmpS5-mmpL5* operons, which encode the Resistance-Nodulation-Division (RND) family efflux pumps (Fig. S2) involved in siderophore secretion[7,10,15]. Rv0455c homologs are highly conserved in mycobacteria (>65% similarity; Fig. S3) and are encoded by genes classified as mycobacterial core genes[18]. To test whether Rv0455c homologs share the same function, we expressed the homologous genes *msmeg_3494* of *M. smegmatis*, *ml2380* and *b586_19750* of the siderophore-deficient species *M. leprae* and *M. haemophilum*, respectively, in the *M. tuberculosis* Δ*rv0455c* deletion mutant using L5 *attB* integration plasmids (Table S2). Interestingly, expression of any of these genes completely restored growth of the *M. tuberculosis* Δ*rv0455c* mutant in the presence of mycobactin (Fig. 2a), suggesting that Rv0455c homologs are functionally equivalent in mycobacteria. Further, we deleted homologous *msmeg_3494* gene in the *M. smegmatis* Δ*fxbA* strain, which is deficient in the production of the exochelin MS siderophores[19,20] (Fig. S4). The Δ*fxbA*/Δ*msmeg_3494* mutant grew like the parent strain in iron-replete medium but had a growth defect in low-iron 7H9 (1 μM $Fe^{3+}$) medium and did not grow in the presence of 1 μM mycobactin, in contrast to the parent strain (Fig. S5). Expression of *rv0455c* from the integration plasmid pML3613 completely restored growth of the Δ*fxbA*/Δ*msmeg_3494* mutant (Fig. S5), indicating that Rv0455c can

substitute for MSMEG_3494 in *M. smegmatis*. Collectively, these results showed that Rv0455c homologs are produced in mycobacteria and that their function in detoxifying mycobactin is conserved.

**Subcellular localization of the Rv0455c protein in *M. tuberculosis*.** Previously, Rv0455c and its homolog were detected in the culture filtrates of *M. tuberculosis* H37Rv[21] and *M. avium* subsp. *avium*[22], respectively, indicating that Rv0455c is secreted by mycobacteria. Bioinformatic analysis of the Rv0455c sequence predicts that the amino acids 1–30 constitute a Sec secretion signal (Fig. S3). Indeed, a truncated gene encoding Rv0455c$_{31-148}$ (Rv0455cΔSP) with a start codon did not rescue the growth defect of the *M. tuberculosis* Δ*rv0455c* mutant in contrast to the intact *rv0455c* gene in low-iron medium containing 100 nM mycobactin (Fig. S6a), indicating that export of Rv0455c is essential for its function. To specifically detect Rv0455c, we used recombinant protein produced in *E. coli* (see Methods) and generated an antiserum. Subcellular fractionation of the avirulent *M. tuberculosis* strain mc$^2$6230 (ΔRD1 ΔpanCD) showed a clear separation of water-soluble versus membrane fractions as indicated by the marker proteins GlpX and EccB5 (Fig. 2b). Quantitative image analysis of equivalent samples separated by gel electrophoresis and detected in Western blots revealed that approximately equal amounts of Rv0455c protein are in the culture filtrate and cell-associated in *M. tuberculosis* mc$^2$6230 (Fig. S7a). To examine

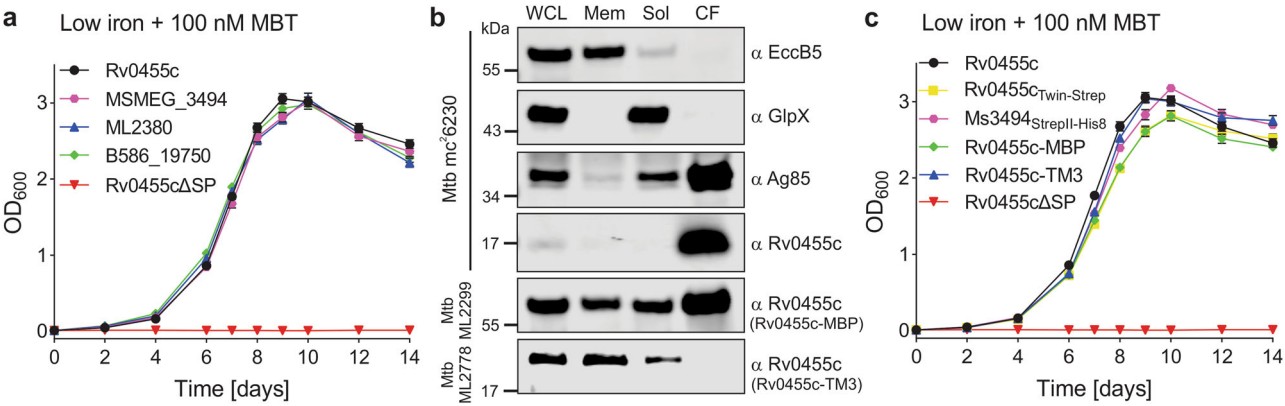

**Fig. 2 Rv0455c is a conserved 14-kDa exported protein. a** Growth curves of the *M. tuberculosis* Δ*rv0455c* mutant complemented with genes encoding the Rv0455c homologs MSMEG_3494 of *M. smegmatis* (*M. tuberculosis* ML2770), ML2380 of *M. leprae* (*M. tuberculosis* ML2294) and B586_19750 of *M. haemophilum* (*M. tuberculosis* ML2295) in 7H9 low iron (<0.1 μM $Fe^{3+}$) medium supplemented with 100 nM Fe-mycobactin (Fe-MBT). The Δ*rv0455c* strains complemented with genes encoding Rv0455c (by *M. tuberculosis* ML2205) and Rv0455cΔSP (Rv0455c$_{31-148}$; by *M. tuberculosis* ML2206) were used as positive and negative controls, respectively. Error bars represent standard deviations from the mean results of biological triplicates (*n* = 3). **b** Subcellular localization of Rv0455c in *M. tuberculosis* mc$^2$6230. Immunoblot analysis of the whole-cell lysate (WCL), the membrane (Mem) and water-soluble fractions (Sol), and of the culture filtrate (CF). The culture filtrate was concentrated 200-fold. Five microliter of each fraction taken from the WCL, Mem2, Sol2, and CF fractions were loaded into 8% SDS-PAGE gels for Western blots. The Rv0455c proteins were detected with an Rv0455c-specific antiserum. EccB5, GlpX, and Ag85 served as indicator proteins for the membrane-associated proteins, soluble proteins, and secreted proteins, respectively. Data were obtained from at least two independent experiments and representative images are shown. **c** Growth curves of the *M. tuberculosis* Δ*rv0455c* mutant (strain ML2203, Table S1) complemented with genes encoding Twin-Strep tagged Rv0455c (ML2298), StrepII-His$_8$ tagged MSMEG_3494 (ML2293), Rv0455c-MBP (ML2299) and Rv0455c-TM3 (ML2778), respectively, in 7H9 low-iron medium (<0.1 μM $Fe^{3+}$) supplemented with 100 nM Fe-mycobactin (Fe-MBT). Error bars represent standard deviations from the mean results of biological triplicates (*n* = 3). Source data are provided in the Source Data file.

whether the ESX-1 system plays a role in this phenotype, we repeated the subcellular fractionation experiments in *M. tuberculosis* mc$^2$6206 (Δ*leuCD* Δ*panCD*) containing all ESX systems (Fig. S7b). Similarly, ~40% of Rv0455c was detected in the culture filtrate of *M. tuberculosis* mc$^2$6206 (Fig. S7c). The phenomenon that proteins with functions in the periplasm are also secreted into the culture filtrate is not uncommon for *M. tuberculosis*. For example, the antigen 85 proteins are mycolyltransferases with their only known functions in the periplasm[23,24], are also found in large quantities in the culture filtrate[25]. In our experiment, ~30% of antigen 85 was detected in the culture filtrate (Fig. 2b).

**Rv0455c performs its essential function in the periplasm of *M. tuberculosis*.** To examine whether secretion of Rv0455c is required for *M. tuberculosis* to grow in the presence of siderophores, we constructed a fusion protein of Rv0455c with the C-terminal maltose-binding protein MBP (Table S2). We assumed that the large Rv0455c-MBP fusion protein (~60 kDa) would be trapped in the periplasm of *M. tuberculosis*, in contrast to the small Rv0455c protein (14 kDa). However, a growth assay in *M. tuberculosis* showed that expression of the *rv0455c-mbp* fusion gene from integration plasmid pML4285 completely rescued the Δ*rv0455c* mutant in low-iron medium with 100 nM mycobactin (Fig. 2c), indicating that the Rv0455c-MBP fusion protein is functional in *M. tuberculosis*. Surprisingly, we still detected ~25% of the Rv0455c-MBP fusion protein in the culture filtrate compared to the *M. tuberculosis* mc$^2$6230 strain producing natural Rv0455c, but significantly more Rv0455c-MBP was cell-associated (Fig. 2b). Next, we constructed an Rv0455c variant containing three C-terminal transmembrane helices to anchor the protein to the membrane and prevent its secretion across the outer membrane. Indeed, subcellular localization experiments showed that the Rv0455c protein with transmembrane helices (Rv0455c-TM3) is mainly localized in the membrane fraction but is not detectable in the culture filtrate (Fig. 2b). Importantly,

expression of the *rv0455c-TM3* fusion gene using the integration plasmid pML4299 (Table S2) completely rescued the Δ*rv0455c* mutant in low-iron medium with 100 nM mycobactin (Fig. 2c) or 1 μM carboxymycobactin (Fig. S6b), demonstrating that (i) membrane-anchored Rv0455c is functional, (ii) that secretion of Rv0455c into the extracellular medium is not required for the function of Rv0455c, and (iii) that the function of Rv0455c is identical for carboxymycobactin and mycobactin. Furthermore, we found that variation of the linker length from 3 aa to 30 aa between Rv0455c and the C-terminal transmembrane helices does not significantly affect the function of Rv0455c (Fig. S6c). In addition, adding the purified proteins, 1 μM Rv0455c$_{31-148}$ (Rv0455cΔSP) or 2 μM MSMEG_3494$_{33-153}$, into the liquid medium failed to restore the growth of the Δ*rv0455c* mutant in the presence of mycobactin (Fig. S6d), suggesting that MSMEG_3494 or Rv0455c does not function outside of the *M. tuberculosis* cells. Collectively, these experiments demonstrated that Rv0455c performs an essential siderophore-related function in the periplasm of *M. tuberculosis*.

**Rv0455c is required for siderophore secretion in *M. tuberculosis*.** Since the toxicity of mycobactins or carboxymycobactins for the *M. tuberculosis* Δ*rv0455c* resembled the siderophore-poisoning phenotype previously observed for the siderophore secretion-defective *M. tuberculosis* Δ*mmpS4*/Δ*mmpS5* mutant[15] (Fig. 1c–f), we hypothesized that Rv0455c is required for siderophore secretion. To examine this hypothesis, mycobactins and carboxymycobactins were radioactively labeled in the parent strain *M. tuberculosis* mc$^2$6230, the Δ*rv0455c*, Δ*mbtD* (no siderophore synthesis[5]) and Δ*mmpS4*/*mmpS5* mutants (no siderophore secretion[7]) by using the radiolabeled precursor salicylic acid as previously described[7]. All strains were grown in self-made low-iron 7H9 medium containing 1 μM ammonium ferric citrate and 16 μM $^{14}$C-salicylic acid for 10 days. The cultures of the Δ*mbtD* mutant were supplemented with 10 μM hemin to enable growth. The

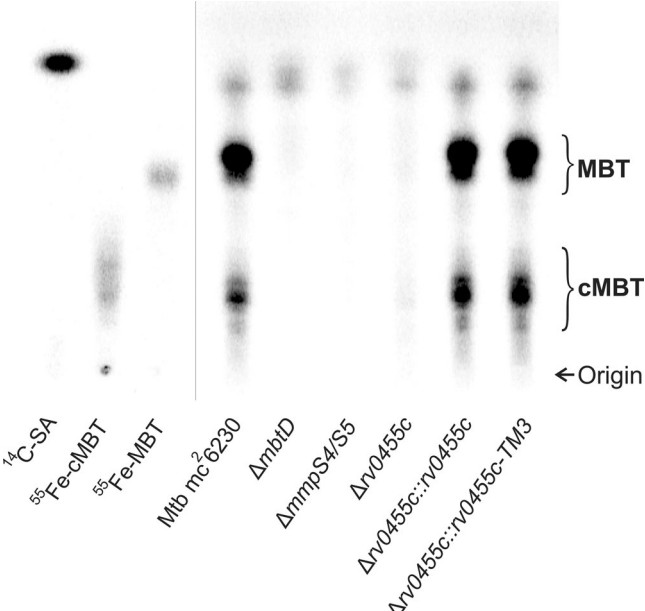

**Fig. 3 Rv0455c is required for siderophore secretion by *M. tuberculosis*.** Thin-layer chromatography of chloroform-extracted siderophores from the culture filtrates of the *M. tuberculosis* mc²6230 parent strain, the siderophore biosynthetic mutant Δ*mbtD*, the siderophore secretion-defective mutant Δ*mmpS4/mmpS5*, the Δ*rv0455c* mutant and the Δ*rv0455c* mutant complemented with integrated *rv0455c* and *rv0455c-TM3* expression vectors. The siderophores were labeled with [7-¹⁴C]-salicylic acid (SA), which was loaded on the silica gel plate as a control together with ⁵⁵Fe-carboxymycobactin (cMBT) and ⁵⁵Fe-mycobactin (MBT). A mixture of ethanol/cyclohexane/water/ethyl acetate/acetic acid with the volume ratios of 5:25:2.5:35:5[42] was used as a solvent. The loaded samples were normalized to their OD₆₀₀ at the time of harvest of the bacterial cultures. Data were obtained from at least two independent experiments and representative images are shown. Source data are provided in the Source Data file.

siderophores were extracted with chloroform from the culture filtrates and analyzed by thin-layer chromatography (TLC). As expected, *M. tuberculosis* mc²6230 secreted mycobactins and carboxymycobactins, which were identified in the TLC by the absence of the respective signals in the siderophore-deficient Δ*mbtD* mutant and by the ⁵⁵Fe-MBT and ⁵⁵Fe-cMBT controls (Fig. 3). Quantitative image analysis revealed that mycobactins in the culture filtrate are the predominant siderophores with a two-fold excess over carboxymycobactins. Importantly, the quantities of both secreted mycobactins and carboxymycobactins were reduced in the Δ*rv0455c* mutant to 4.9% and 9.6%, respectively, compared to the parent *M. tuberculosis* mc²6230 strain (Fig. 3). This result is very similar to that obtained for the Δ*mmpS4/S5* mutant, which showed a reduction of mycobactin and carboxymycobactin levels to 1.4% and 0.7%, respectively, compared to the parent strain (Fig. 3). Complementation of the Δ*rv0455c* mutant by wild-type Rv0455c and the membrane-anchored Rv0455c (Rv0455c-TM3) fully restored siderophore secretion to wild-type levels (Fig. 3). These results demonstrated that Rv0455c is required for secretion of both mycobactins and carboxymycobactins by *M. tuberculosis*.

### The chemical diversity and structures of siderophores produced by *M. tuberculosis* are not altered by deletion of the *rv0455c* gene

To examine whether siderophore synthesis is altered in the Δ*rv0455c* mutant, the lipidomes of *M. tuberculosis* mc²6230, the Δ*rv0455c* mutant and the complemented mutant

were analyzed. To obtain siderophore quantities sufficient for lipidomics, all strains were grown in 15-fold larger volumes of low-iron 7H9 medium compared to the TLC experiments. Lipids were extracted with chloroform/methanol from the cell pellets and analyzed as previously described[15]. Mycobactins typically have long-chain fatty acids (C18-C20), whereas carboxymycobactins are rendered more soluble by their short-chain (C5-C12) and negatively charged dicarboxylates[26]. Both siderophores also exhibit different amino acid modifications and altered hydroxylation and saturation states, which affect their biological activities[12,27] (Fig. 4a). We generated molecular species by collision-induced dissociation mass spectrometry (CID-MS) and tracked them based on mass differences of the mycobactic acid unit (Fig. S8). This fragment reveals the saturation state of the fatty acids and lengths of carboxymycobactins (C5-C12) and mycobactins (C18-C20) (Fig. 4b, c). This method also generates a fragment ion, which distinguishes methylserine and serine modifications (Fig. S8c). High-performance liquid chromatography combined with time-of-flight mass spectrometry (HPLC-TOF-MS) revealed the presence of all major ferri-siderophore species in lipid extracts of both the parent *M. tuberculosis* mc²6230 strain and the Δ*rv0455c* mutant (Fig. 4b, c) demonstrating that the loss of Rv0455c does not change the hydroxylation state, methylserine use and fatty acyl length, and saturation of the detected siderophores. Importantly, lipidomics revealed an identical chemical diversity of the siderophores produced by the parent strain *M. tuberculosis* and the Δ*rv0455c* mutant, strongly indicating that the loss of Rv0455c does not alter the siderophore synthesis in *M. tuberculosis*. The signals of the total cell-associated mycobactins in the Δ*rv0455c* mutant were reduced by ~40% compared to the parent *M. tuberculosis* mc²6230 strain, while the total cell-associated carboxymycobactins were increased by four-fold in the Δ*rv0455c* mutant (Fig. 4d). Importantly, similar quantitative changes of the cell-associated siderophores were observed in previous lipidomics experiments for the siderophore secretion-deficient Δ*mmpS4/S5* mutant and were explained by the observed intracellular accumulation of the normally secreted hydrophilic carboxymycobactins[15]. In conclusion, the lipidomics experiments demonstrate that the lack of Rv0455c does not change the chemical diversity and structures of the siderophores synthesized by *M. tuberculosis*. Thus, the observed phenotypes of the Δ*rv0455c* mutant are indeed caused by the impaired secretion of siderophores and the subsequent downregulation of siderophore production in *M. tuberculosis*.

### The DUF5078 domain of Rv0455c exhibits a novel structure

To determine the structure of Rv0455c, we produced the Rv0455c₃₁₋₁₄₈ (Rv0455cΔSP) protein with a C-terminal hexahistidine-tag. However, this protein accumulated in inclusion bodies. Based on the observation that mutation of either of the two conserved cysteines (Fig. S3) in Rv0455c completely abolished the function of Rv0455c (Fig. S6e), we hypothesized that these cysteines form an intramolecular disulfide bond. To enable disulfide bond formation in the periplasm, we fused the Rv0455c₃₁₋₁₄₈ (Rv0455cΔSP) protein with the signal peptide of OmpF, an exported porin of *E. coli*. Export into the *E. coli* periplasm significantly increased the amount of soluble Rv0455c protein, but the purified protein rapidly aggregated at concentrations exceeding 50 μM. To circumvent these difficulties, we examined the close homolog MSMEG_3494 of *M. smegmatis*, which shares 66% similar amino acids with Rv0455c (Fig. S3) and has the same function (Fig. 2a). Production of MSMEG_3494₃₃₋₁₅₃ with a C-terminal StrepII-His₈ tag in the periplasm of *E. coli* yielded a large amount of soluble protein (Fig. S9a). Growth experiments

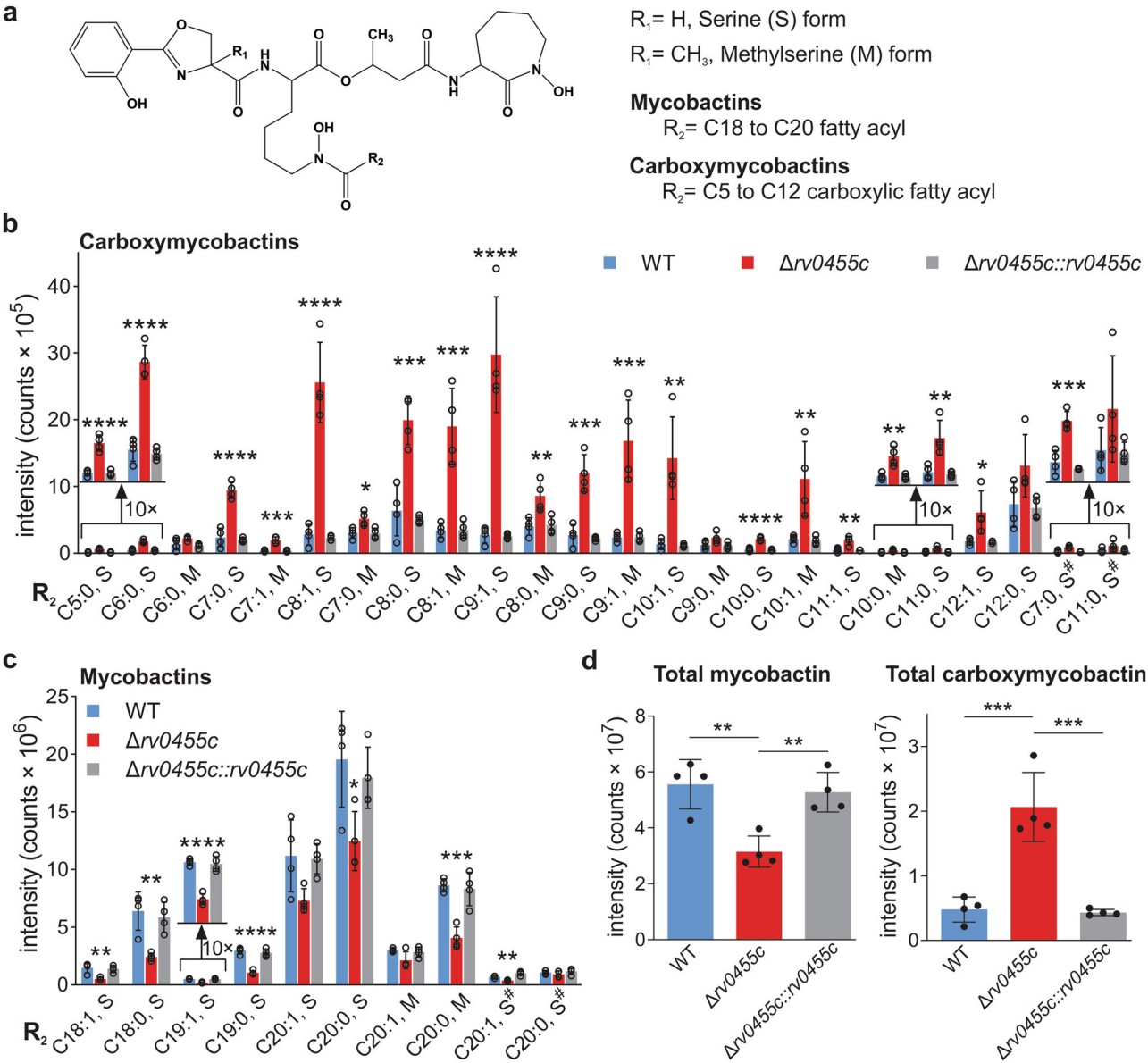

**Fig. 4 Siderophore profile of the *M. tuberculosis* Δ*rv0455c* mutant.** Whole-cell lipid extracts of *M. tuberculosis* mc²6230, the Δ*rv0455c* mutant, and the Δ*rv0455c* mutant complemented with an integrated *rv0455c* expression vector were analyzed by lipidomics using high-performance liquid chromatography and mass spectroscopy. **a** Chemical structure of the core of *M. tuberculosis* siderophores. $R_1$: H or $CH_3$ represent the serine (S) or methylserine (M) form, respectively. $R_2$: alkyl chains. The mean intensity values for molecular events are shown for a subset of carboxymycobactin isoforms (**b**) and mycobactin isoforms (**c**). The indicated fatty acid chain lengths (number of carbons) and the number of double bonds corresponding to mono- or di-carboxyl units assume the peptide structure shown in (**a**), which according to prior analyses is the dominant form[12,55]. The monodeoxy-mycobactins and monodeoxy-carboxymycobactins are indicated with a hash tag (#). **d** Total mycobactin and total carboxymycobactin counts in wt *M. tuberculosis* (blue), the Δ*rv0455c* mutant ML2203 (red), and the complemented strain ML2205 (gray). Data are mean ± s.d. of four biological replicates. Asterisks indicate significant differences (*$p < 0.05$, **$p < 0.01$, ***$p < 0.001$, ****$p < 0.0001$ by one-way ANOVA with Tukey's test) compared with wt *M. tuberculosis*. Source data are provided in the Source Data file.

with *M. tuberculosis* using the integrative expression vectors pML4279 and pML4284 (Table S2) showed that MSMEG_3494 with a C-terminal StrepII-His$_8$ tag and Rv0455c with a C-terminal Twin-Strep tag fully restored the growth of the *M. tuberculosis* Δ*rv0455c* mutant in medium with mycobactin (Fig. 2c), indicating that small C-terminal tags do not affect the functions of either MSMEG_3494 or Rv0455c in *M. tuberculosis*. The MSMEG_3494$_{33–153}$ protein was purified by Ni(II)-affinity chromatography, and its affinity tag was removed by enterokinase (Fig. S9b). The MSMEG_3494$_{33–153}$ protein was further purified by size exclusion chromatography yielding a single peak with an estimated

molecular weight of 15 kDa consistent with its predicted value indicating that the MSMEG_3494$_{33–153}$ protein is a monomer (Fig. S9a). The MSMEG_3494$_{33–153}$ protein was soluble at concentration of 1 mM. The three-dimensional structure of MSMEG_3494$_{33–153}$ was solved by X-ray crystallography and refined to 2.1 Å (Table S4). As no structures of MSMEG_3494$_{33–153}$ homologs are known, experimental methods were required to determine the phases instead of molecular replacement methods. To this end, the crystal was soaked with sodium bromide, and the phases were determined by multiwave anomalous dispersion (MAD) using the bromide ions as the source of anomalous scattering. The unit cell contained

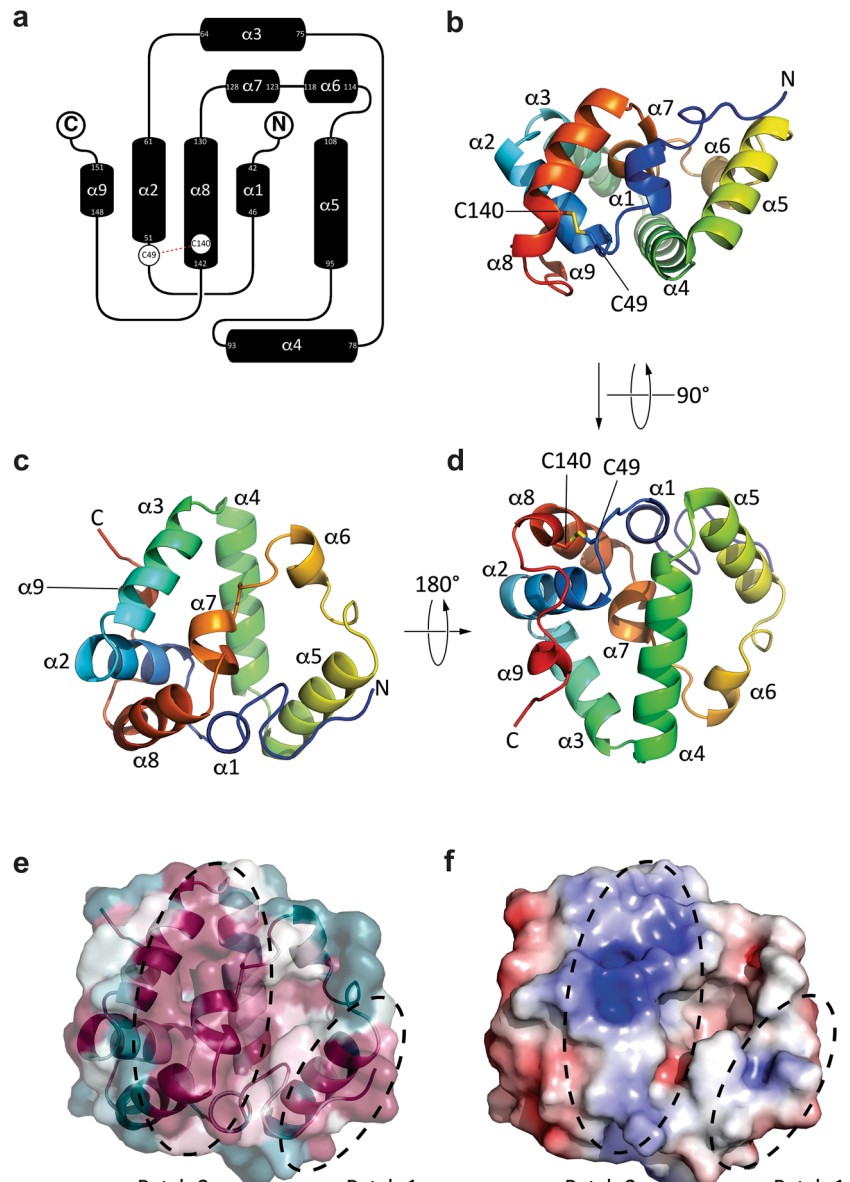

**Fig. 5 Structure of the Rv0455c homolog MSMEG_3494. a** Topology of MSMEG_3494$_{33-153}$. N- and C-termini are indicated, helices are depicted as black tubes numbered from N-terminus to C-terminus (α1-α9; other numbers represent the sequence number for the first and last residue of the helix). The disulfide bond between C49 and C140 is indicated with a red dashed line. Loops are not drawn to scale. **b–d** Cartoon depictions of MSMEG_3494$_{33-153}$ colored by amino acid sequence from the N-terminus (blue) to the C-terminus (red). The disulfide bond between C49 and C140 is shown using stick representation and colored yellow. Spatial relationship between the orientations in **b–d** are reflected by the rotational arrows between each image. **e** Colors reflect the conservation of amino acids across the 150 closest members of DUF5078 to MSMEG_3494. Purple indicates high conservation, white indicates moderate conservation, and teal indicates a lack of conservation. Patches 1 and 2 are highlighted within black dashed ellipses. The orientation is the same as in (**c**). **f** Surface depiction of the electrostatic surface potential from -5 kT/e (red) to +5 kT/e (blue). Patches 1 and 2 are highlighted within black dashed ellipses. The orientation is the same as in (**c**).

two molecules of MSMEG_3494$_{33-153}$ and 11 bromide ions with ~42% of the volume being occupied by solvent molecules. MSMEG_3494$_{33-153}$ adopts a novel structure consisting of a helical bundle with four single turn α-helices and five longer α-helices (Figs. 4a–d, S10a). A disulfide bond between C49 and C140 connects the N-terminus of helix 2 and the C-terminus of helix 8, establishing a cinch-like topology (Fig. 5a). These two cysteines are conserved across all members of the DUF5078 homology (Fig. S3), indicating that the disulfide bond plays an important role in structure stabilization. In addition, we found that other evolutionarily conserved amino acids clustered predominantly in two

distinct surface-exposed patches (Figs. 4e, S10b, c): the interface between the flexible N-terminus and helix 5 (patch 1), and the interfacial region of helices 2, 3, 7, and 8, (patch 2). Analysis of the electrostatic potential map of MSMEG_3494$_{33-153}$ revealed a highly electropositive surface for patch 2, which may form an interaction interface accommodating proteins or molecules with negatively charged domains or moieties (Figs. 4f, S10d). Patch 1 did not display a significant number of charges, but the residues constituting patch 1 exhibited large *B*-factors indicating conformational flexibility and the potential for adopting structures in solution different from that in the crystal lattice (Fig. S10e).

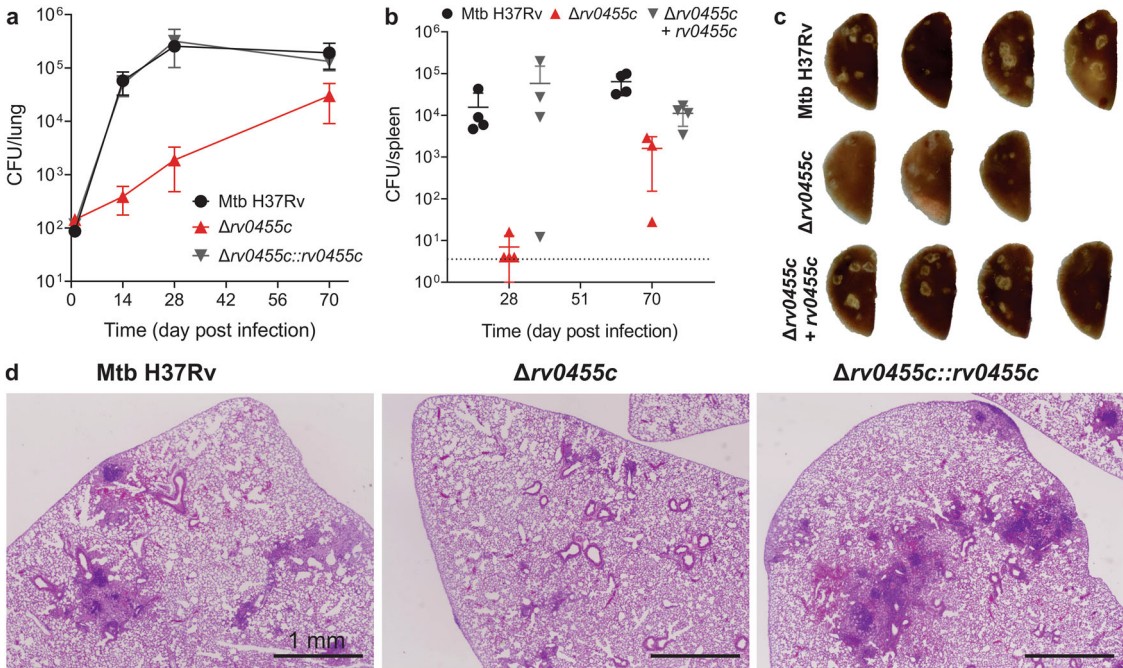

**Fig. 6 Rv0455c is a virulence factor of *M. tuberculosis* in mice.** C57BL/6 mice infected with the WT *M. tuberculosis* parent strain (ML2710), the Δ*rv0455c* mutant (ML2711), and the complemented strain (ML2701). Colony-forming unit (CFU) counts in the lungs (**a**) and spleens (**b**) of mice infected with the indicated strains. Data represent the mean ± s.d. of four mice per time point. Note that one mouse infected with Δ*rv0455c* mutant was euthanized on day 51 due to an unrelated illness. Source data are provided in the Source Data file. **c** Gross pathology scans of left lung lobes of mice harvested on day 70 after infection. **d** Histopathology of lungs of mice harvested on day 70 after infection. The scale bar is 1 mm. Data were obtained from two independent experiments and representative images are shown.

**The MSMEG_3494 protein does not interact with carboxymycobactins in vitro.** It is unclear how Rv0455c mediates siderophore secretion. It is possible that periplasmic Rv0455c has an essential structural role in the siderophore secretion system of *M. tuberculosis*, e.g., as an adapter protein as AcrA in the tripartite efflux system of *E. coli*[28]. An alternative hypothesis is that Rv0455c acts as a periplasmic carrier of siderophores and contributes to the transport of siderophores to the outer membrane. Such a function would require siderophore binding. To test whether Rv0455c homologs bind siderophores, we conducted 2D $^1$H-$^{15}$N HSQC NMR titrations and examined the chemical shift perturbations of $^{15}$N-labeled MSMEG_3494$_{33-153}$ in the presence of deferrated carboxymycobactin S from *M. smegmatis*. No significant difference in the spectra was observed (Fig. S11). To examine the unlikely possibility that Rv0455c might bind iron-loaded carboxymycobactin, we added gallium (III), a commonly used substitute to avoid interference from paramagnetic iron (III)[29], at a ratio of 1:1 to deferrated carboxymycobactin. However, we did not observe any changes of the chemical shift perturbations of the $^{15}$N-labeled MSMEG_3494$_{33-153}$ protein. These experiments indicate that the recombinant MSMEG_3494 protein does not interact with carboxymycobactin in vitro.

**Rv0455c is required for full virulence of *M. tuberculosis* in mice.** To assess the role of Rv0455c for virulence of *M. tuberculosis*, C57BL/6 mice were infected with low dose aerosols containing the *M. tuberculosis* H37Rv parent strain (ML2710), the Δ*rv0455c* mutant (ML2711) and the complemented strain (ML2701). Loss of the *rv0455c* gene severely compromised the ability of *M. tuberculosis* to proliferate in lungs and in spleens as the number of viable bacteria decreased by 140- and 2200-fold, respectively, compared to the parent *M. tuberculosis* H37Rv on day 28 post infection (Fig. 6a, b). The differences were reduced to 6- and 40-fold in lungs and spleens, respectively, on day 70 post infection (Fig. 6a, b). The

expression of *rv0455c* from the mycobacteriophage L5 integration plasmid pML3613 completely restored the ability of the Δ*rv0455c* mutant to proliferate in lungs and spleens. These results demonstrate that Rv0455c is necessary for *M. tuberculosis* to establish a lung infection in mice, especially in the early stage. Gross pathological examination and histological assessment on lungs of mice infected with either the parent strain *M. tuberculosis* H37Rv or the *rv0455c* complemented strain exhibited extensive lesions and displayed significant lymphocytic infiltrates on day 70 post infection (Fig. 6c, d). In contrast, lungs of the mice infected with the Δ*rv0455c* deletion mutant showed many fewer lesions or much weaker lymphocytic infiltrates (Fig. 6c, d). In conclusion, the infection experiments showed that Rv0455c is required for full virulence of *M. tuberculosis* in mice.

## Discussion

**Rv0455c is essential for siderophore secretion in *M. tuberculosis*.** Rv0455c is one of the 219 mycobacterial core proteins, which are shared between *M. tuberculosis* and *M. leprae*, have no homologs outside of mycobacteria and are implicated to be involved in pathways unique to mycobacteria[18]. In this study, we showed that siderophore secretion by *M. tuberculosis* is strongly reduced in the *rv0455c* deletion mutant, while the chemical diversity and structures of the detected siderophores are identical to that of wild-type *M. tuberculosis*. These results and other phenotypes of the *rv0455c* deletion mutant such as growth defect in low-iron medium even in the presence of heme and toxicity of siderophores, are very similar to those observed for the *M. tuberculosis* siderophore secretion mutant lacking the MmpS4 and MmpS5 proteins[7,15]. Mechanistically, these phenotypes are explained by the accumulation of both exogenous mycobactins and carboxymycobactins in the Δ*mmpS4/mmpS5* mutant compared with the parent *M. tuberculosis* strain as observed previously in uptake experiments[15]. As a consequence, siderophore secretion mutants reduce siderophore

production to avoid the toxic accumulation of siderophores inside the cells, as observed also for enterobactin secretion-deficient *E. coli* mutants[13,30]. This mechanism might involve the inhibition of the (carboxy)mycobactin biosynthesis complex assembly as proposed for the glycopeptidolipid secretion-deficient *M. smegmatis mmpS4* deletion mutant[31]. The toxicity of external siderophores for *M. tuberculosis* mutants with a defective siderophore secretion system in the presence of heme was termed siderophore poisoning[15] and is based on siderophore recycling: the repetitive cycle of uptake of iron-loaded siderophores, removal of iron from siderophores by a reductive mechanism and secretion of empty siderophores[15]. Thus, all phenotypes of the *M. tuberculosis rv0455c* deletion mutant are consistent with impaired siderophore secretion and recycling.

The proteins which are targeted by empty siderophores trapped within the *M. tuberculosis* cell are unknown, but may include the essential iron-sulfur cluster proteins involved in bacterial energy production, such as the NADH dehydrogenase complex I[32]. The potential of the mainly membrane-associated, hydrophobic MBTs[6] to target membrane proteins may explain the 10-20-fold increased toxicity of mycobactin compared to cMBTs for *M. tuberculosis* Δ*rv0455c* mutant (Fig. 1e, f) and for other siderophore secretion mutants of *M. tuberculosis*[15]. Taken together, in this study, we identified Rv0455c as the first single protein essential for siderophore secretion by *M. tuberculosis*. It remains unclear how it interacts with the two siderophore export pumps consisting of the MmpL4/MmpS4 and MmpL5/MmpS5 proteins[7,15].

**Rv0455c is essential for virulence of *M. tuberculosis* in mice**. In this study, we showed that Rv0455c is the first single protein involved in siderophore secretion which is required for virulence of *M. tuberculosis* in mice. While the Δ*mmpS4/mmpS5* mutant also has a virulence defect[7], two genes need to be deleted since the MmpS4/MmpL4 and MmpS5/MmpL5 efflux systems have similar functions in siderophore secretion. Although these proteins function in the same pathway, interesting differences in their roles in virulence were evident. While the bacterial burden in the lungs of mice 28 days after infection with the *M. tuberculosis* Δ*rv0455c* deletion mutant was 140-fold reduced compared to wt *M. tuberculosis*, it was >10,000 fold reduced for the *mmpS4/ mmpS5* deletion mutant compared to the parent *M. tuberculosis* H37Rv strain[7]. The larger attenuation of the *mmpS4/mmpS5* double deletion mutant is consistent with the increased toxicity of mycobactin compared to the *rv0455c* deletion mutant (Fig. 1e), indicating that self-poisoning of the *M. tuberculosis* mutants with impaired siderophore secretion is the likely cause of their virulence loss. Our results also indicate that the *rv0455c* deletion mutant is more permissive for residual siderophore secretion than the *mmpS4/mmpS5* deletion mutant. This conclusion is consistent with the slow but significant replication of the *M. tuberculosis rv0455c* deletion mutant in the lungs of mice (Fig. 6a, c), while the *mmpS4/mmpS5* deletion mutant did not replicate at all and showed no signs of lung pathology[7] in contrast to *M. tuberculosis* Δ*rv0455c*. Taken together, our studies provide a proof of principle that single proteins required for siderophore secretion exist and further validate siderophore secretion as a suitable target for novel tuberculosis drugs.

**Structure and function of Rv0455c**. Our finding that Rv0455c is secreted by *M. tuberculosis* is in accordance with previous proteomic analyses[21,33], but was difficult to reconcile with its essential function in siderophore secretion. One possible mechanism is that mature Rv0455c binds empty siderophores in the periplasm and subsequently transports siderophores across the outer membrane. Similar mechanisms were reported for the

transport of mycobacterial lipids such as phthiocerol dimycocerosates (PDIM) and triacylglycerides (TAG) by the periplasmic lipoproteins LppX[34] and LprG[35], respectively, to the outer membrane. However, the lack of chemical shift perturbations by NMR spectroscopy indicated that the close Rv0455c homolog MSMEG_3494 does not interact with carboxymycobactin. This finding is consistent with the absence of a deep cleft in the crystal structure of MSMEG_3494 (Fig. S10) and in the identical structure of Rv0455c (Fig. S12) predicted by AlphaFold[36]. Deep binding pockets are commonly found in siderophore-binding proteins (Fig. S13) and in the lipid-binding LppX and LprG proteins (Fig. S14). Our structural analysis showed that MSMEG_3494 and Rv0455c consist of nine α-helices, while LprG and LppX consist of mainly β-sheets (Fig. S14a, b) and possess large and hydrophobic cavities to accommodate their lipid substrates (Fig. S14c, d). A narrow and deep hydrophobic groove capable of binding mycobactin was also observed for CD1a (Fig. S15), a protein which is part of the antigen-presenting CD1 complex in mammalian cells[37]. These structural comparisons show that the DUF5078 family proteins are different from the LprAFG family or other proteins and may not directly bind small lipophilic molecules such as mycobactin and carboxymycobactin. However, we cannot exclude that our experimental conditions were not suitable for carboxymycobactin binding by MSMEG_3494 or that MSMEG_3494 might require another protein or protein modifications for siderophore binding. For example, it is conceivable that Rv0455c is capable of carboxymycobactin binding after folding in the periplasm and then switches to a closed configuration with the formation of the essential disulfide bond. Similar protein dynamics have been reported for azurin, a copper-binding protein from *Pseudomonas aeruginosa*[38].

On the other hand, the observation that membrane-anchored Rv0455c-TM3 is fully functional in the Δ*rv0455c* mutant rules out a function of secreted Rv0455c in siderophore secretion. Instead, these results support an alternative mechanism in which Rv0455c plays an important structural role in the siderophore secretion system of *M. tuberculosis*, possibly by acting as an essential accessory protein to maintain the activity of the MmpL efflux systems or by connecting the inner membrane siderophore exporters MmpL4/MmpS4 and MmpL5/MmpS5[7] with a putative outer membrane channel (Fig. 7). The hypothesis of a functional connection of Rv0455c with the siderophore efflux systems is supported by the close proximity of the *rv0455c* orthologous genes to *mmpL4* or *mmpL5* in all mycobacterial species (Fig. S2) and by our structural analysis revealing two surface patches, which might provide docking interfaces for other proteins (Fig. 5e, f). An example of such a mechanism is the AcrB-AcrA-TolC tripartite efflux system, which mediates siderophore secretion in many Gram-negative bacteria. In *E. coli* the inner membrane RND efflux pump AcrB and outer membrane channel TolC are connected by the periplasmic adapter protein AcrA[13]. The multi-functionality of the AcrB-AcrA-TolC tripartite efflux system in siderophore secretion and drug efflux would also explain why even the siderophore-deficient species *M. leprae* and *M. haemophilum* still possess the fully functional Rv0455c homologs ML2380 and B586_19750.

Our study identifies the essential function of the mycobacterial core protein Rv0455c in siderophore secretion and in virulence of *M. tuberculosis* in mice. Rv0455c is exported across the inner membrane by the SecYEG translocon, and likely plays a structural role in siderophore transport to the outer membrane (Fig. 7). This study presents a major step forward in understanding the *M. tuberculosis* siderophore secretion system, whose proteins have no similarities to other bacterial siderophore secretion systems[14,39–41]. Furthermore, we identify siderophore poisoning

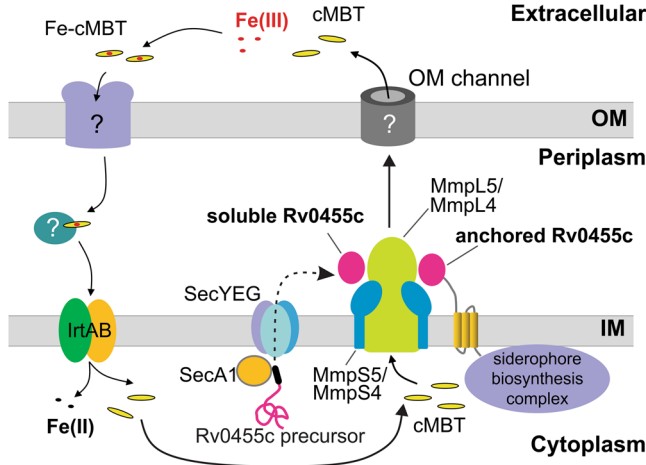

**Fig. 7 Model of siderophore secretion and recycling in *M. tuberculosis*.**
Iron-loaded siderophores are taken up on the cell surface by a putative siderophore receptor and then transferred to the inner membrane importer IrtAB[56–58]. The transport across the periplasmic space is probably mediated by proteins of unknown identity. Then, ferric siderophores are taken up by IrtAB across the inner membrane (IM). During this process, iron (III) is reduced and dissociates from the siderophores. Deferrated siderophores are exported again, along with the export of de novo synthesized siderophores, by a process dependent on the MmpL4/S4 and MmpL5/S5 efflux systems[7,10,15]. The Rv0455c pre-protein is translocated across the inner membrane via SecA1 secretion system. The mature, water-soluble Rv0455c protein is located in the periplasm and is essential for secretion of carboxymycobactin (cMBT), possibly by interaction with the MmpL4/5-MmpS4/5 systems. Transport of siderophores across the outer membrane (OM) requires at least one hitherto unknown protein.

as an important mechanism of the large virulence loss of the *M. tuberculosis* mutant lacking the *rv0455c* gene, validating siderophore secretion as a drug target and revealing a new mechanism for putative tuberculosis drugs.

## Methods

**Bacterial strains, reagents, media, and growth conditions**. All the strains used in this study are listed in the Table S1. The avirulent *M. tuberculosis* strain mc²6230 (H37Rv ΔRD1 ΔpanCD) and its derivative strains were grown in Middlebrook 7H9 broth (Difco) supplemented with 10% ADS (50 g/L bovine albumin, 20 g/L dextrose, 8.5 g/L NaCl), 0.5% glycerol, 24 μg/mL pantothenate, 0.2% casamino acids and 0.01% tyloxapol. The virulent *M. tuberculosis* H37Rv and its derivative strains were grown in Middlebrook 7H9 broth (Difco) supplemented with 10% OADC (Middlebrook oleic acid-albumin-dextrose-catalase), 0.5% glycerol, 0.2% casamino acids and 0.01% tyloxapol. *Escherichia coli* strains were grown in either LB medium or M9 minimal medium containing appropriate antibiotics at 37 °C with shaking at 200 rpm. The following antibiotics were used when required: kanamycin (Kan) at 30 μg/mL for mycobacteria or *E. coli*, and hygromycin (Hyg) at 200 μg/mL for *E. coli*, and 50 μg/mL for mycobacteria. Purified iron-free and iron-loaded carboxymycobactins (cMBTs) from *M. smegmatis* were purchased from EMC Microcollections (Germany). Iron-loaded mycobactin (MBTs) was purified from *M. smegmatis* by chromatography[42] and was provided by Dr. Colin Ratledge. All other chemicals were purchased from Sigma at the highest quality available.

**Construction of *M. tuberculosis* gene deletion mutants and complementation strains**. The *rv0455c*, *fxbA*, and *msmeg_3494* genes deletion were performed by homologous recombination strategy using previous procedure[43]. The parent *M. tuberculosis* mc²6230 strain harboring the *rv0455c* deletion vector pML3694 were grown on the selective 7H10 plates containing 10% ADS, 0.5% glycerol, 24 μg/mL pantothenate, 0.2% casamino acids, 20 μM hemin, 2% (w/v) sucrose and 50 μg/mL hygromycin. The plates were cultured at 40 °C for 4 weeks to select for the double cross-over (DCO) colonies which were then validated by PCR (Fig. S1a, b). The *rv0455c* gene deletion in the virulent strain *M. tuberculosis* H37Rv was done by the same protocol and was validated by PCR (Fig. S1a, c). For *fxbA* deletion, a suicide vector pML3655 was transformed into *M. smegmatis* mc²155 to create the single cross-over (SCO) colonies on 7H10 plates supplemented with 50 μg/mL hygromycin. Then, 2% sucrose was used to select for the DCO colonies, which were then validated by PCR (Fig. S4a, b). The unmarked strain Δ*fxbA::loxP* (ML2249) was

created by the *cre* expressing vector pML2714. Then, *msmeg_3494* deletion in the ML2249 strain was done with the same strategy by a suicide vector pML3618. The Δ*fxbA::loxP* Δ*msmeg_3494::hyg*ʳ (ML2275) double mutant was validated by PCR (Fig. S4c, d). A series of expression vectors of *rv0455c* or *msmeg_3494* were transformed into either *M. tuberculosis* Δ*rv0455c* (ML2203) or *Msm* Δ*fxbA* Δ*msmeg_3494* (ML2275) strains to integrate into the chromosomal L5 attB sites for generating the complementation strains (Table S1). All the mutants are listed in Table S1, and the deletion and expression vectors are listed in Table S2.

**Growth assays**. The *M. tuberculosis* mc²6230 strain (as wt) and its derivative strains were pre-grown in 7H9/hemin medium (supplements: 0.5% glycerol, 24 μg/mL pantothenate, 0.2% casamino acids and 0.01% tyloxapol) until the $OD_{600}$ reached ~1.0. Afterward, the cells were grown in self-made low-iron 7H9 medium (<0.1 μM $Fe^{3+}$ as determined by ICP-MS) containing the supplements for 5–7 days to deplete intracellular iron sources. The iron-depleted cells were inoculated into 10 mL of low-iron 7H9 medium (containing the supplements) at an initial $OD_{600}$ of 0.01 with different iron sources: 10 μM hemin; 10 μM hemin plus 100 nM Fe-mycobactin; 10 μM hemin plus 1 μM Fe-carboxymycobactin, respectively. The $OD_{600}$ of the cultures were determined every 24 or 48 h. All experiments were done in triplicate.

**Siderophore toxicity assays**. The siderophore toxicity assays were performed as previously described[15]. Purified mycobactins (Fe-MBTs) and carboxymycobactins (Fe-cMBTs) dissolved in ethanol were serially diluted using 7H9 medium with supplements (10% ADS, 20 μM Hemin, 0.5% glycerol, 24 μg/mL pantothenate, 0.2% casamino acids and 0.01% tyloxapol) in 96-well plates in 100 μL volumes by triplicate. The pre-grown *M. tuberculosis* cultures using 7H9 medium containing the supplements with an $OD_{600}$ of ~0.5 were diluted to the $OD_{600}$ of 0.02. One hundred microliter volumes of each cell suspension were distributed to all wells receiving cells. Plates were incubated for 5–6 days at 37 °C with 230 rpm shaking on a microplate mixer. Forty μL of alamarBlue working solution (alamarBlue:10% Tween-80 = 1:1) was added and the plates were incubated at 37 °C with 230 rpm for 8 h. The fluorescence of each well was analyzed using a Biotek Synergy HT plate reader with a 530 nm excitation and a 590 nm emission filter. The growth inhibition assays % viability was determined by subtracting background fluorescence from un-inoculated samples followed by normalizing fluorescence values to samples that did not receive any treatment. Fluorescence values from untreated cells were set at 100%.

**Thin-layer chromatography of siderophores from *M. tuberculosis* culture filtrates**. Radiolabeling of siderophores was performed in a similar manner as previously described with modifications[7]. The *M. tuberculosis* strains mc²6230 (parent strain), Δ*mbtD* (ML1600), Δ*mmpS4/S5* (ML859), Δ*rv0455c* (ML2203), Δ*rv0455c::rv0455c* (ML2205) and Δ*rv0455c::rv0455c-TM3* (ML2778) (Table S1) were pre-grown in 10 mL Middlebrook 7H9 medium containing 10 μM hemin, 10% ADS, 0.2% glycerol, 24 μg/mL pantothenate, 0.2% casamino acids and 0.02% tyloxapol to $OD_{600}$ of ~1.5. Then, the cells were washed with phosphate buffered saline (PBS; 137 mM NaCl, 2.7 mM KCl, 2 mM $KH_2PO_4$, 8 mM $Na_2HPO_4$, pH 7.4) and transferred to 20 mL self-made 7H9 medium without added iron (total iron concentration <0.1 μM as determined by inductively coupled plasma mass spectrometry[10]) containing supplements (0.2% glycerol, 24 μg/mL pantothenate, 0.2% casamino acids and 0.02% tyloxapol) for two more days to deplete intracellular iron stores. Subsequently, the iron-depleted *M. tuberculosis* strains were inoculated into 8 mL of low-iron 7H9 medium containing the supplements as described above with 1 μM ammonium ferric citrate and 16 μM [7-¹⁴C]-salicylic acid (0.75 μCi/mL, American Radiolabeled Chemicals) were grown for 10 days with shaking at 37 °C. The cultures of the Δ*mbtD* mutant were supplemented with 10 μM hemin to enable growth. The cultures of the WT, the *rv0455c* complementation strains, and the Δ*mbtD* mutant reached $OD_{600}$ of 2.5 from an initial $OD_{600}$ of 0.15. The cultures of the Δ*mmpS4/S5* and the Δ*rv0455c* mutants reached $OD_{600}$ of 1.3 and 2.2, respectively, from an initial $OD_{600}$ of 0.6. The cultures were centrifuged for 10 min at 5000 × g and the supernatants were collected. The supernatants were filtered through a 0.22 μm filter (Millipore). Ferric chloride (100 mM $FeCl_3$ in ethanol) was added at a final concentration of 0.6 mM to saturate siderophores in the supernatants with iron at room temperature for 1 h. Then, the supernatants were extracted twice with 5 mL chloroform and the organic fractions were pooled and retained. The chloroform extracts of the supernatants were then evaporated using a Vacufuge (Eppendorf) and resuspended in 100–200 μL of chloroform to normalize to the final $OD_{600}$ of the cultures. The extracts were then subjected to thin-layer chromatography on silica gel 60 plates (10 × 10 cm, 250 μm thick; Sigma) using ethanol/cyclohexane/water/ethyl acetate/acetic acid (5:25:2.5:35:5) as a solvent as described previously[42]. The ⁵⁵Fe-loaded mycobactins and carboxymycobactins, as well as radiolabeled salicylic acid substrate were used as controls. Plates were allowed to dry and subjected to autoradiography for 72 h. The phosphor screen was imaged using a Typhoon Trio+ Imager (GE).

**Lipidomics**. Total lipid extraction of *M. tuberculosis* was performed as previously described[15] with modifications. *M. tuberculosis* mc²6230 (parent strain), Δ*rv0455c*

(ML2203), and complementation strain (ML2205) were pre-grown in 20 mL self-made 7H9 medium without additional iron (<0.1 μM Fe) for 1 week to deplete intracellular iron stores. Cells were harvested and washed twice with PBS buffer. Afterward, the cells were grown in 120 mL detergent-free, low-iron 7H9 medium (supplemented with 1 μM ammonium ferric citrate, 0.5% glycerol, 24 μg/mL pantothenate, and 0.2% casamino acids) at an initial $OD_{600}$ of 0.04 for 8–11 days. The cells were harvested after the cultures reached an $OD_{600}$ of ~2. Approximately 1 g of the cell pellets were extracted two times with 15 mL $CHCl_3$:MeOH (1:2) for 2 h and 6 mL $CHCl_3$:MeOH (1:1) for 2 h, respectively. The total lipid extracts were dried in the vacuum. Lipid masses per sample were measured, and dried lipids were resuspended to 1 mg/mL in $CHCl_3$:MeOH (1:1). Fifty microgram of total lipid per technical and biological run were injected for HPLC/MS analyses. All experiments were done in quadruplicate. Mass-normalized samples of the lipid extracts containing MBTs and cMBTs were analyzed using an Agilent 6520 accurate-mass quadrupole time-of-flight mass spectrometer with an electrospray source and an Agilent 1200 series HPLC system with a Poroshell C18 column (3 mm × 50 mm). The elution solvent was a mixture of methanol:water (A) (7:3) and n-propanol:cyclohexane (9:1) with 2 mM ammonium formate (B). Solvent flow was carried out at a rate of 0.15 mL/min. A gradient from 0% B (0–4 min) with a linear gradient (4–13 min) to 100% B (13–22 min) was used. Mycobactin and carboxymycobactin annotations were made based on 10 ppm mass match of ions with the indicated core polyketide-peptide[12] and deduced acyl chain length of the mono- or di-carboxylate unit.

**Protein expression and purification.** The C-terminal StrepII-His$_8$ tagged MSMEG_3494$_{33–153}$ was expressed in *E.coli* BL21 (DE3) by periplasmic expression vectors pML4249 or pML4604 containing a *E.coli* signal sequence (Table S2). One liter of Luria–Bertani (LB) medium was used for producing the unlabeled protein, and two liters of M9 minimal medium containing 4 g $^{15}NH_4Cl$ were used for producing $^{15}N$-labeled proteins. Cells were grown at 37 °C until the cell density reached an $OD_{600}$ of ~1.8. Then the cells were induced with 1 mM isopropyl-β-D-thiogalactopyranoside (IPTG) at 18 °C overnight. Afterward, cells were harvested by centrifugation and lysed by sonication in ice-cold lysis buffer (150 mM NaCl, 30 mM Tris, 1 mM PMSF [pH 7.5]). Cell lysate was clarified by centrifugation, and the supernatant flowed through Ni-NTA column. Approximately 20 mg of the protein was obtained by Ni affinity purification. Afterward, the affinity tag was removed by enterokinase, and a total of ~15 mg isotopically labeled protein was obtained after size exclusion chromatography via Superdex 75 gel filtration column (GE).

**Carboxymycobactin-binding experiments.** NMR experiments were collected at 25 °C on a Bruker Avance II (700 MHz $^1H$) spectrometer equipped with a cryogenic triple-resonance probe, processed with NMRPIPE[44], and analyzed with NMRVIEW[45]. $^{15}N$-labeled protein samples were prepared at 50 μM in a buffer containing 50 mM sodium phosphates [pH 6.5]. NMR titrations were performed with 20 mM stock solution of iron-free carboxymycobactin S (EMC company, Germany) dissolved in 100% DMSO. carboxymycobactin S was mixed with the StrepII-His$_8$ tagged MSMEG_3494$_{33–153}$ protein at a molar ratio of 2:1. The mixture was incubated at 37 °C for 1 h, followed by acquisition of 2D $^1H$-$^{15}N$ HSQC data.

**Crystallization of the MSMEG_3494 protein.** After enterokinase cleavage and subsequent purification as described above, a solution of MSMEG_3494$_{33–153}$ (1 mg/mL) was dialyzed overnight into a 20 mM Tris-HCl buffer (pH 7.75) at 4 °C. After centrifugation to clarify the solution, the protein was concentrated to 8 mg/mL using an Amicon Ultra 10 KDa MWCO centrifugal filter. Initial crystallization trials were conducted in 96-well sitting drop plates at both 4 °C and room temperature with either a 1:1 or 2:1 ratio of protein to precipitant. Crystals with a hexagonal prism geometry were observed in well F1 of the JBScreen Kinase HTS precipitant screen, containing 0.2 M LiAc and 28% PEG 4 K. For further optimization, quality crystals for data collection were grown at room temperature after mixing 2 μL of a solution containing 8 mg/mL MSMEG_3493$_{33–153}$ in a 20 mM Tris-HCl buffer with a pH of 7.75 with 1 μL of precipitant solution (0.2 M LiAc, 26% PEG 4 K) and equilibration in a hanging drop plate with 300 μL of precipitation solution. To obtain heavy atom derivative, the crystals were soaked with 1.5 M NaBr in the cryoprotective buffer containing 0.2 M LiAc, 26% PEG 4 K, and 20% glycerol for 180 sec before flash freezing in liquid nitrogen.

**X-ray data acquisition and structure determination.** MAD data from the NaBr soaked crystals were collected at the Stanford Synchrotron Radiation Lightsource (SSRL) using beam line 9–2. Nine hundred frames with an oscillation width of 0.2° were collected for each wavelength of the experiment. SHELX C/D/E[46] was used to identify positions of 11 bromide ions. 115 out of 127 residues for each chain were built automatically with Buccaneer from CCP4i2[47]. The model was completed and refined to 2.1 Å resolution using COOT[48] and REFMAC5[49]. In the final model, residues 155–158 of chain A and residues 32, 33, 156–158 of chain B were disordered. Residues 32 and 154–158 were artifacts of the expression and purification processes used. The data quality and refinement statistics are presented in Table S4. The structure coordinates and data have been deposited in the PDB (PDB ID: 7REF).

**Structure validation and analysis.** The conformation and nonbonded interaction parameters were evaluated using the program MolProbity[50]. Structures were rendered, examined, and visualized with PyMol (Molecular Graphics System, Version 2.5, Schrödinger, LLC). Electrostatic potentials were calculated in units of kT/e, where k is the Boltzmann constant, T is the absolute temperature, and e is the proton charge, using the programs PDB2PQR[51] and APBS[52]. Evolutionarily conserved residues were identified using the ConSurf server, with the percentage sequence identity range spanning from 35 to 95% during multiple sequence alignment of closest 150 sequences[53].

**Generation of Rv0455c antiserum.** The C-terminal hexahistidine tagged Rv0455c$_{31–148}$ (Rv0455cΔSP) was produced in *E.coli* BL21 (DE3) using the periplasmic expression vector pML4212 containing a signal sequence from *E.coli* OmpF. After Ni(II)-affinity purification, 10 mg purified protein was loaded on 8% SDS-PAGE gels for separation. The recombinant Rv0455c protein bands were cut from the gels. Approximately 5 mg of the protein was extracted and used for producing the polyclonal antiserum in rabbits (GenScript).

**Subcellular localization for Rv0455c and its variants in *M. tuberculosis*.** The subcellular localization experiments were performed using previous procedure[54]. The *M. tuberculosis* strains expressing the original Rv0455c or the Rv0455c variants were grown in 500 mL 7H9 [10 μM $Fe^{3+}$] medium till $OD_{600}$ reached ~0.6. The cell pellet and supernatant were separated by centrifugation at 3000 g for 10 min. To obtain culture filtrate (CF) fraction, the supernatant was filtered through 0.22 μm Millipore filtration system and then concentrated ~200-fold to a final volume of 2.5 mL using a 3-kDa-cutoff Amicon (Millipore) ultrafiltration device. The cell pellet was resuspended in 10 mL PBS buffer containing 1 mM phenylmethylsulfonyl (PMSF) and lysed by sonication (24 w output power, 15 min "ON" in total). Cell debris was removed from the lysate by centrifugation at 3200 × g for 10 min at 4 °C. The 10 mL clear lysate (WCL) was centrifuged at 100,000 × g for 1 h at 4 °C. The supernatant (Sol1) was transferred to a separate tube, and the pellet (Mem1) was resuspended in another 10 mL PBS buffer. Afterward, both S1 and M1 fractions were centrifuged at 100,000 × g for 1 h at 4 °C. The supernatant (Sol2) containing the cytosolic fraction was transferred to a new tube. The membrane pellet (Mem2) was resuspended in 10 mL PBS with 1% SDS. Finally, 5 μL of each fraction taken from the WCL, Mem2, Sol2, and CF were analyzed by 8% SDS-PAGE gels and Western blots, as shown in Fig. 2b and S7b. The membrane protein EccB5, the cytosolic protein GlpX, and the culture filtrate proteins Ag85 and CFP10 were detected by rabbit polyclonal anti-EccB5 antibody (1:1000), anti-GlpX antibody (1:1000), anti-Ag85 antibody (1:5000, BEI Resources) and anti-CFP10 antibody (1:5000, BEI Resources), respectively, as controls. The Rv0455c and its variants were detected by the rabbit polyclonal anti-Rv0455c antibody (1:300). To analyze the proportion of the secreted Rv0455c by *M. tuberculosis*, 20 μL of the whole-cell lysate and 5 μL of the culture filtrate (2.5 mL) were loaded into 8% SDS-PAGE gels for Western blot. The secreted Rv0455c versus the cell-associated Rv0455c is ~1:1 in *M. tuberculosis* mc$^2$6230 (Fig. S7a) and ~2:3 in *M. tuberculosis* mc$^2$6206 (Fig. S7c). Quantitative image analysis of western blots was done using ImageJ 1.46r.

**Mouse infection experiments.** The phthiocerol dimycocerosates (PDIM) of the strains for mice experiments was confirmed by thin-layer chromatography (Fig. S1h) using previous procedure[43]. We utilized an inhalation exposure system (Glas-Col) to infect 7–8 weeks old female C57BL/6 mice (Jackson Laboratory) with the *M. tuberculosis* strains (ML2710, ML2711, and ML2701) which are listed in Table S1. Cultures used for infections were in mid-log phase of growth and prepared such that each mouse received ~200 bacilli. At the indicated time points, mice were euthanized by carbon dioxide inhalation. Lungs and spleens were isolated and homogenized in PBS using a bullet-blender (Next Advance). Dilutions of the homogenates were cultured on 7H10 agar supplemented with 20 μM hemin (Sigma-Aldrich) to determine the number of bacteria (CFU). On day 28 and day 70 post infection, the left upper lung lobes were preserved in formalin. The preserved tissues were embedded in paraffin wax, sliced, and stained with hematoxylin and eosin at the Weill Cornell Laboratory of Comparative Pathology. The mice experiments were repeated once.

**Ethics statement.** All animal experiments were performed following National Institutes of Health guidelines for housing and care of laboratory animals and performed in accordance with institutional regulations after protocol review and approval by the Institutional Animal Care and Use Committee of Weill Cornell Medical College (Protocol Number 0601441 A).

**Reporting summary.** Further information on research design is available in the Nature Research Reporting Summary linked to this article.

## Data availability

Atomic coordinates of the MSMEG_3494 crystal structure have been deposited in the Protein Data Bank under accession number 7REF. The data generated in this study are provided in the Supplementary Information and Source Data files. Additional data are available upon request. Source data are provided with this paper.

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

## Acknowledgements

We thank Dr. Carolina Trujillo for help with the mouse infection experiments, Dr. Todd Green for initial crystallization experiments, Drs. Virginia Meikle and David Pajuelo-Gamez for assistance in the BSL-3 laboratory, and Drs. Mikhail Pavlenok and Uday Tak for help with protein purification. We thank Dr. Colin Ratledge for purified siderophores and Dr. Wilbert Bitter for the anti-EccB5 antibody. We thank the O'Neal Comprehensive Cancer Center at the University of Alabama at Birmingham (funded by the NCI grant P30 CA013148) for supporting the High-Field NMR facility. This work was supported by the National Institutes of Health grants R01 AI049313 to D.B.M. and R21 AI151239 to M.N.

## Author contributions

M.N. and L.Z. conceived the study. L.Z. constructed the plasmids and strains, performed growth assays and in vitro experiments, and purified proteins. J.E.K. and A.E.A. produced protein crystals and solved the Ms3494 structure. M.W., D.C.Y., and D.H. performed animal infection, lipidomics and NMR experiments, respectively. Y.K. did initial crystallization experiments. L.Z., M.W., J.E.K., F.M.M., S.E., G.C. J.S.S., D.B.M., and M.N. designed experiments. M.N., S.E., F.M.M., J.S.S., D.B.M., and G.C. directed research. All authors analyzed data and edited the manuscript. L.Z. and M.N. wrote the manuscript with input from all authors.

## Competing interests

The authors declare no competing interests.
