## [Peer Review File · Nature Communications]

Reviewers' Comments:

Reviewer #1:

Remarks to the Author:

The manuscript investigates the function of Rv0455c in iron acquisition and virulence. The authors resolve the structure of this protein and show it is important for growth in conditions of iron limitation and during mice infection. These are significant results. However, the functional mechanism of Rv0455c remains unclear. The proposed function in the periplasm does not agree with the natural localization of the protein in the extracellular environment. The data is well presented, but many details in the methodology need clarification, some controls are lacking, and additional experiments are needed to support the main conclusions.

Specific Comments:

Introduction:

The introduction correctly states that unlike carboxymycobactin, mycobactin is a membrane associated siderophore. The following statement in line 33 "it is unclear how MBTs and cMBTs are transported through the periplasm and across the outer membrane" contradicts the previous statement about mycobactin.

Furthermore, in the title and in various sentences through the manuscript, the authors postulate a role of rv0455 in siderophore secretion. Carboxymycobactin secretion is more appropriate since mycobactin remains membrane associated.

Methods:

1. Because non virulent and virulent H37Rv strains are used in this study, the authors should be very clear when the rv4455c mutant derived from the attenuated strain (which it is a mutant itself) or the virulent strain are used. The phenotypes of the mutant derived from the attenuated strain should be reproduced in the virulent strain used for mice infection.

2. In the siderophore toxicity assays what medium (iron content) is used to dilute the cells?

3. Inhibition of Alamar blue reduction is an indicator of bacteriostasis not necessarily cell killing therefore, it is not a viability assay. CFUs should be enumerated to evaluate cell viability.

4. Different Fe or heme concentrations are used in various experiments without justification.

5. According to the methods in the lipidomic section, the mutant depleted of iron grows like the wild type from 0.04 to 2.0 O.D in medium containing 1uM Fe, which is still very low. This is in contrast with the results shown in Fig 1 where the mutant does not grow at all in low iron. A titration of iron required by the mutant for growth will be very helpful to interpret the results.

6. It should be specified whether the mycobactin and carboxymycobactin used in the siderophore toxicity assays are Fe bound or deferrated. Is it the siderophore or the Fe-siderophore complex responsible for the toxicity observed?

Results:

7. Because mycobactin is not normally secreted, it is difficult to understand how lack of Rv0455c leads to the toxic effect of mycobactin particularly when there is no accumulation of cellular mycobactin in the mutant. In fact, less mycobactin is found in the mutant cells. Thus, the premise of hypersensitivity of rv4455c to mycobactin is unclear.

8. Despite that it is unclear how non-functional rv4455c results in mycobactin toxicity, the authors use complementation of mycobactin hypersensitivity as a proxy for functionality of rv4455c homologs as well as the membrane anchored form of the protein in siderophore secretion. To support the function of rv0455c in carboxymycobactin secretion, in the periplasm, genetic complementation of carboxymycobactin secretion by membrane attached rv4455c is needed.

7. The authors refer to a common mechanism of mycobactin and carboxymycobactin in toxicity as 'siderophore poisoning' but there is no evidence of mycobactin accumulation in the mutant. Thus, it seems that the effect of the rv0455c mutation is different for mycobactin and carboxymycobactin. This should be clarified in the results section.

8. The evidence for impaired carboxymycobactin secretion in the mutant is weak. The ratio of cellular versus extracellular carboxymycobactin in mutant and WT should be compared to demonstrate reduced secretion in the mutant. Deregulation of carboxymycobactin synthesis could account for the increased cellular level observed in the mutant.

9. The authors show that naturally produced rv0455c is extracellular which is very difficult to reconcile with the proposed function in the periplasm. It also raises the strong possibility that the secreted protein is indeed the contributor to Mtb virulence. According to the authors' model, a fraction of Rv0455c functions in the periplasm in carboxymycobactin secretion. This conclusion is based on complementation of the mycobactin hypersensitivity phenotype by the artificially

membrane attached protein. To support their model the authors should test the ability of the membrane attached form of the protein to complement both the postulated carboxymycobactin secretion defect as well as the virulence phenotype.

8 . Genes involved in iron acquisition are generally regulated by iron. Is rv4455c expression responsive to iron? Does it localize in the extracellular medium versus the periplasm in an iron-dependent manner? Perhaps it is retained in the periplasm by proteins that are synthesized in low iron (like the mmpL/mmpS proteins). Co-IP experiments that test the interaction of Rv0455c with MmpL4/5-Mmps4-5 would support the structural role proposed for Rv0455c in the periplasm and would explain the functional localization. Having already generated a specific antibody against Rv0455c co-IP experiments are not difficult.

11. Did the authors test the rv04455c mutant derived from virulent Mtb in other stressful conditions other than iron limitation?

Reviewer #2:

Remarks to the Author:

The manuscript by Zhang et al describes the role of Rv0455c in Mtb siderophore export and virulence. The investigators also solve the structure of this core mycobacterial protein.

The authors have previously shown a role for MmpL4/MmpL5 system for secretion of siderophores. The Rv0455c gene is within the same genomic region as MmpL4. A previously published transposon mutant screen suggests that Rv0455c was required for Mtb growth in mycobactin and showed a similar siderophore poisoning phenotype as the mmpS4/5 mutants. Based on this, a clean deletion mutant was made in the avirulent and virulent Mtb strains. The rv0455c deletion strain failed to grow in low iron or media containing mycobactin similar to the mmpS4/5 mutant. Slower growth was also observed in medium containing hemin as the only iron source, similar to the mycobactin mutant. Mycobactin toxicity could be heterologously complemented by other mycobacterial homologues, which is really nice to see and supports a conserved mechanism for this core mycobacterial protein. The authors further demonstrate that Rv0455c is secreted to the periplasmic space – and this localization is required for function as shown by deletion of signal peptide, a TM-fusion construct. Further, addition of exogenous protein does not rescue the mutant. The role for siderophore secretion was demonstrated through mass spectrometry, further supporting the model of siderophore-poisoning. Despite this, the purified protein does not appear to bind mycobactins. The structure of the MSMEG3494 homologue was determined, and shows a unique structure with a cysteine “cinch”. Finally, the mutant was attenuated compared to wild type in a mouse model of virulence.

Overall, the manuscript is well-written, the data clearly presented and interpreted. These results represent a significant advance to the field. The authors have placed their current findings in the context of their previous studies on MmpL4/5 systems. A remaining question is whether Rv0455c interacts with MmpS4/5 or periplasmic domains of MmpL4/5. These experiments are legitimately a lot of work, but if such studies have been attempted it may warrant a mention in the discussion.

I had a couple of minor suggestions to help the reader follow the story:

Supplemental figure 1, can you add the S 15 here to show conservation and genomic context in a broader fashion.

I think the authors should decide between Rv0455SP or Rv045531-148 and use consistently.

Reviewer #3:

Remarks to the Author:

This manuscript from Zhang et al provides novel functional insights into the mycobactin siderophore secretion pathway in Mycobacteria which involves a previously uncharacterized protein Rv0455c. The authors use a wide array of microbiological, biochemical and in vivo experiments to characterise the function of this protein. Genetic knock-out of this protein results in siderophore poisoning, which is also observed for known siderophore secretion genes such as mmpS4/S5. These careful genetic knock-out experiments, together with lipidomics the authors show that the

protein is involved in secretion of mycobactins. However, no binding was detected between apo siderophores and purified Rv0455c as seen by NMR spectroscopy. The authors also went on to determine the structure of a homologue of Rv0455c from *M. smegmatis* (also functional in *Mtb*) using X-ray crystallography, the model revealed a novel protein fold, however offered a limited mechanistic detail about the functional mechanism. The biochemical and structural aspects of the paper are done very well. The data collection statistics and model refinement seem to be fine for the resolution.

Overall the paper contains a wealth of information about siderophore secretion in a very important pathogen *M. tuberculosis*. The *in vivo* experiments solidify the importance of siderophore secretion pathways as targets for developing antimicrobial compounds against *Mycobacteria*. I think this paper will appeal to a wide audience and I recommend its publication after addressing some of my comments.

1) I have a hard time understanding the secretion versus function in the periplasm aspect of the paper. The authors show that majority of the protein is secreted into the media, yet only functional in the periplasm. The model shown in figure 6 does not include this aspect of its function, even though it's unclear presently. They also show that secretion of the protein into the media is not necessary for its function. Could the protein in media be a result of cell lysis?

2) Have the authors checked whether the homologues in *M. smegmatis* and *M. abscessus* are also secreted into culture media?

3) Out of curiosity, have the authors tried to see if the protein can interact with the surface of liposomes using NMR? Perhaps it needs to be immobilized onto a lipid surface for association.

Minor comments:

Have the authors performed structure similarity searches with the DALI web server? They describe the structure as a novel fold, but it would be good to specify whether there are any distant structural homologues.

Are the two proteins in the ASU identical? Perhaps specify rmsd somewhere (either in the methods section or in a figure legend)

Page 7, lines 140, what TM helices were used for the fusion construct? I could not find the info anywhere.

Page 10 lines 214: specify that the structure was solved by X-ray crystallography.

Page 21, line 493: SHELX not SHELLX, please fix

Rebuttal – Manuscript NCOMMS-21-31817

Manuscript title: A periplasmic cinched protein is required for siderophore secretion and virulence of *Mycobacterium tuberculosis*

Authors: Zhang et al.

Reviewer 1

The manuscript investigates the function of Rv0455c in iron acquisition and virulence. The authors resolve the structure of this protein and show it is important for growth in conditions of iron limitation and during mice infection. These are significant results. However, the functional mechanism of Rv0455c remains unclear. The proposed function in the periplasm does not agree with the natural localization of the protein in the extracellular environment. The data is well presented, but many details in the methodology need clarification, some controls are lacking, and additional experiments are needed to support the main conclusions.

We agree that we do not know the molecular mechanism how Rv0455c exerts its function in siderophore secretion. This will be an exciting project in the future. The discovery of the function of Rv0455c and its structure as described in this manuscript will provide the basis for these follow-up studies. We addressed the specific critiques including the subcellular localization as follows.

Introduction:

The introduction correctly states that unlike carboxymycobactin, mycobactin is a membrane associated siderophore. The following statement in line 33 “it is unclear how MBTs and cMBTs are transported through the periplasm and across the outer membrane” contradicts the previous statement about mycobactin.

There is no contradiction. Mycobactins are hydrophobic and, hence, associated with membranes, but they still need to be transported across the outer membrane and secreted in order to function as siderophores. It has been shown many times including in this study that mycobactins alone are sufficient to supply *M. tuberculosis* with external iron. A short statement has been added to clarify that (l. 26-32).

Furthermore, in the title and in various sentences through the manuscript, the authors postulate a role of rv0455 in siderophore secretion. Carboxymycobactin secretion is more appropriate since mycobactin remains membrane associated.

The reviewer is not correct. Both exogenous mycobactin and carboxymycobactin rescue the siderophore biosynthesis mutant $\Delta mbtD$ (*PLoS Pathog* **16**:e1008337; 2020), demonstrating that mycobactin and carboxymycobactin function as siderophores in iron acquisition and **are secreted** for this purpose. This information has been added to the introduction (lines 29-32). Additionally, both siderophores inhibit the RND efflux pump mutants $\Delta mmpS4/S5$ or $\Delta mmpL4/L5$ (*PNAS* **111**:1945-50; 2014 & *PLoS Pathog* **16**:e1008337; 2020), and the $\Delta rv0455c$ mutant in this study, indicating that mycobactin and carboxymycobactin share the same secretion pathway in *Mtb*.

Methods:

1. Because non virulent and virulent H37Rv strains are used in this study, the authors should be very clear when the rv0455c mutant derived from the attenuated strain (which it is a mutant itself) or the

virulent strain are used. The phenotypes of the mutant derived from the attenuated strain should be reproduced in the virulent strain used for mice infection.

In this study, we deleted *rv0455c* gene from the attenuated Mtb mc²6230 (H37Rv Δ RD1 Δ panCD) and from the virulent strain Mtb H37Rv for the mouse experiments. To address the reviewer's concern, we used the *rv0455c* deletion mutant derived from virulent Mtb H37Rv for growth experiments and showed that the siderophore poisoning phenotype is identical with that of the *rv0455c* deletion mutant from Mtb mc²6230 (new Figs. S1e-g). This is now described in lines 84-87. We also made sure that the different Mtb strains are clearly described throughout the manuscript.

2. In the siderophore toxicity assays what medium (iron content) is used to dilute the cells?

For the siderophore toxicity assays we used the standard 7H9 Middlebrook medium [150 μ M Fe³⁺] (Difco) and supplemented it with 20 μ M hemin, 10% ADS, 0.5% glycerol, 24 μ g/mL pantothenate, 0.2% casamino acids and 0.01% tyloxapol for culturing the Mtb cells. We diluted the cells using the same medium. This was described in the methods section before. Now, we added to the methods section that the 7H9 medium used for the dilutions also contained the supplements (l. 446). Under iron-replete conditions, wt Mtb and the complemented strains achieve optimal growth and the micromolar quantities of iron from iron-loaded siderophores or hemin or the iron traces in water and medium components are negligible.

3. Inhibition of Alamar blue reduction is an indicator of bacteriostasis not necessarily cell killing therefore, it is not a viability assay. CFUs should be enumerated to evaluate cell viability.

We deleted this sentence. Now we only describe the dose-dependent "growth inhibition" (line 90). Since we already determined that siderophore poisoning is bactericidal for Mtb (PNAS 111:1945-50; 2014), we did not repeat this experiment for the Δ *rv0455c* mutant.

4. Different Fe or heme concentrations are used in various experiments without justification.

The iron salt and hemin concentrations used in this work were developed and published in our previous studies about iron acquisition by Mtb (PNAS 111:1945-50; 2014; PLoS Pathog 16:e1008337; 2020). These papers are cited in the results (e.g. in line 84) and methods sections (e.g. in line 444).

5. According to the methods in the lipidomic section, the mutant depleted of iron grows like the wild type from 0.04 to 2.0 O.D in medium containing 1 μ M Fe, which is still very low. This is in contrast with the results shown in Fig 1 where the mutant does not grow at all in low iron. A titration of iron required by the mutant for growth will be very helpful to interpret the results.

The reviewer is correct that the Δ *rv0455c* mutant grows slowly at low Fe³⁺ concentrations (1-10 μ M). To account for this growth defect, we increased the initial cell density (OD₆₀₀ to 0.04) when the culture of the Δ *rv0455c* mutant was inoculated. Importantly, the cultures for all Mtb strains were harvested at the same OD₆₀₀ ~2.0 for the lipidomics experiment. To further address the reviewer's concern, we have performed a growth assay with the Mtb mc²6230, Δ *rv0455c* and Δ *mmpS4/S5* strains at an initial OD₆₀₀ of 0.01 in the low iron (< 0.1 μ M Fe³⁺) 7H9 medium supplemented with different concentrations of ferric ammonium citrate (1 μ M, 10 μ M, 100 μ M). The Δ *mmpS4/S5* mutant does not grow at all in low iron (1-10 μ M), while the Δ *rv0455c* mutant still grows slowly at those conditions. These results are described (lines 71-75) and shown in the new Fig. S1d.

6. It should be specified whether the mycobactin and carboxymycobactin used in the siderophore toxicity assays are Fe bound or deferrated. Is it the siderophore or the Fe-siderophore complex responsible for the toxicity observed?

The iron-load forms of mycobactin and carboxymycobactin are used in the siderophore toxicity assays and all the growth assays also. We performed these experiments according to the previous methods (*PNAS* **111**:1945-50;2014). The forms of the siderophores are specified in the Methods section (line 445) and the related figure legends (lines 620, 625). Using the iron-free siderophores or the iron-loaded siderophore would not affect the results since the iron-replete medium was used in the siderophore toxicity assays.

Results:

7. Because mycobactin is not normally secreted, it is difficult to understand how lack of Rv0455c leads to the toxic effect of mycobactin particularly when there is no accumulation of cellular mycobactin in the mutant. In fact, less mycobactin is found in the mutant cells. Thus, the premise of hypersensitivity of rv0455c to mycobactin is unclear.

We understand the reviewer's confusion regarding the function of mycobactin, since nobody knows why *M. tuberculosis* has two different classes of siderophores (carboxymycobactin and mycobactin). This is probably related to functions outside of Mtb, when Mtb is in a cellular environment. However, **current studies unequivocally show that both mycobactins and carboxymycobactins are secreted and have the same function in iron acquisition**. A summary of the current status of the literature was added to the manuscript: "While MBTs are mainly membrane-associated and cMBTs are secreted, both siderophores are detected in the culture filtrate of Mtb^{7,8} demonstrating that not only carboxymycobactins but also mycobactins are secreted. This is consistent with the observation that lipid vesicles secreted by Mtb contain mycobactin⁹. Importantly, addition of mycobactin or carboxymycobactin to the culture medium rescues the growth of an Mtb mycobactin synthesis mutant in low-iron medium demonstrating that both classes of siderophores are functionally identical in iron acquisition¹⁰." (lines 26-31). Furthermore, many experiments in our study also show that the functions of mycobactins and carboxymycobactins in iron acquisition and their mechanism of secretion are identical and consistent with previously observations (*PNAS* **111**:1945-50;2014).

8. Despite that it is unclear how non-functional rv0455c results in mycobactin toxicity, the authors use complementation of mycobactin hypersensitivity as a proxy for functionality of rv0455c homologs as well as the membrane anchored form of the protein in siderophore secretion. To support the function of rv0455c in carboxymycobactin secretion, in the periplasm, genetic complementation of carboxymycobactin secretion by membrane attached rv0455c is needed.

The experiments demonstrating that mycobactin and carboxymycobactin are both secreted and functionally identical in iron acquisition are described in the introduction (lines 26-31) as described above. To address the reviewer's concern regarding the periplasmic localization of Rv0455c, we performed a growth assay with the $\Delta rv0455c$ mutant complemented with membrane attached Rv0455c and showed that **carboxymycobactin completely rescues the $\Delta rv0455c$ mutant in low-iron medium (new Fig. S6b)** as previously shown for mycobactin (line 165). This result also demonstrates that the function of Rv0455c is identical for carboxymycobactin and mycobactin (lines 167- 168).

9. The authors refer to a common mechanism of mycobactin and carboxymycobactin toxicity as 'siderophore poisoning' but there is no evidence of mycobactin accumulation in the mutant. Thus, it seems that the effect of the *rv0455c* mutation is different for mycobactin and carboxymycobactin. This should be clarified in the results section.

We stated above and now also in the manuscript that there is no detectable difference in the functions of mycobactin and carboxymycobactin in iron acquisition by *Mtb*. The reviewer is correct that there is no accumulation of mycobactin in the *rv0455c* deletion mutant. This phenomenon was also observed for other siderophore secretion mutants such as the *mmpS4/S5* mutant. This is likely due to a feedback inhibition of mycobactin biosynthesis when export is blocked (*PNAS* **111**:1945-50;2014). This likely explanation is now mentioned in the results section (lines 205-208). A similar feedback mechanism was previously observed for the *mmpS4* mutant in *M. smegmatis* for glycopeptidolipids (*Mol Microbiol* **78**, 989-1003; 2010). This statement was also added to manuscript (lines 211-212).

10. The evidence for impaired carboxymycobactin secretion in the mutant is weak. The ratio of cellular versus extracellular carboxymycobactin in mutant and WT should be compared to demonstrate reduced secretion in the mutant. Deregulation of carboxymycobactin synthesis could account for the increased cellular level observed in the mutant.

Carboxymycobactins accumulated in the Δ *rv0455c* mutant by a factor of four compared to the parent strain. This difference is very significant as shown in Fig. 3. A similar phenotype was observed previously for the Δ *mmpS4/S5* mutant (*PNAS* **111**:1945-50;2014). We also showed previously that the Δ *mmpS4/S5* mutant does not secrete any mycobactins and only very small quantities of carboxymycobactins (*PLoS Pathog* **9**, e1003120; 2013). The lipidomics experiment further showed that the total cell associated siderophores (mycobactins plus carboxymycobactins) from the Δ *rv0455c* mutant are almost identical (~90%) with those in the WT (Fig. 3b), indicating that siderophore biosynthesis is not dysregulated in the Δ *rv0455c* mutant (Lines 212-215). This statement was added to the manuscript.

11. The authors show that naturally produced *rv0455c* is extracellular which is very difficult to reconcile with the proposed function in the periplasm. It also raises the strong possibility that the secreted protein is indeed the contributor to *Mtb* virulence. According to the authors' model, a fraction of Rv0455c functions in the periplasm in carboxymycobactin secretion. This conclusion is based on complementation of the mycobactin hypersensitivity phenotype by the artificially membrane attached protein. To support their model the authors should test the ability of the membrane attached form of the protein to complement both the postulated carboxymycobactin secretion defect as well as the virulence phenotype.

We agree with the reviewer that it is confusing that Rv0455c is both in the periplasm of *Mtb* and secreted. However, our data convincingly show that membrane-anchored Rv0455c is sufficient to restore growth of the Δ *rv0455c* mutant in medium containing mycobactin (Fig. 2c). As requested by the reviewer we did the same experiment with carboxymycobactin and showed that also cMBT restores wt growth of the Δ *rv0455c* mutant (new Fig. S6b). This new experiment is also described in the results section (line 165).

In addition, we determined the relative quantities of Rv0455c in the periplasm versus the culture filtrate. In contrast to the superficial impression from the Western blots due to the concentration of the culture filtrate, quantitative image analysis and accounting for the different concentration factors showed that approximately equal amounts of Rv0455c protein are in the culture filtrate and cell-associated in *Mtb*

mc²6230. This is now described in the text (lines 136-142). Furthermore, the phenomenon that proteins with functions in the periplasm are also secreted into the culture filtrate is not uncommon for Mtb. For example, the antigen 85 proteins are mycolyltransferases with their only known functions in the periplasm^{23,24}, are also found in large quantities in the culture filtrate²⁵. This statement was added to the text (lines 142-146).

Since there is no reasonable doubt that the function of Rv0455c is in the periplasm we did not do the requested the very elaborate and expensive lipidomics and mouse infection experiments.

12. Genes involved in iron acquisition are generally regulated by iron. Is rv0455c expression responsive to iron? Does it localize in the extracellular medium versus the periplasm in an iron-dependent manner? Perhaps it is retained in the periplasm by proteins that are synthesized in low iron (like the mmpL/mmpS proteins). Co-IP experiments that test the interaction of Rv0455c with MmpL4/5-MmpS4-5 would support the structural role proposed for Rv0455c in the periplasm and would explain the functional localization. Having already generated a specific antibody against Rv0455c co-IP experiments are not difficult.

The *rv0455c* gene is not iron-regulated according to the several transcriptome studies of Mtb under iron-limited conditions (*Infect Immun* **70**:3371-81; 2002 & *Microbiology* **153**:1435-1444; 2007). 10 μ M iron was included in the self-made 7H9 medium for Mtb growth for subcellular localization experiments (see Methods, lines 550-552). This iron concentration stimulates expression of most of the iron-responsive genes such as *mmpL4/S4* and *mmpL5/S5* in Mtb (*Infect Immun* **70**:3371-81; 2002 & *Microbiology* **153**:1435-1444; 2007).

We agree with the reviewer that it would be interesting to examine putative protein interactions of Rv0455c by co-immunoprecipitation as suggested or by crosslinking. However, these experiments would also require extensive follow-up experiments to eliminate false-positive results and to confirm putative interactions *in vitro*. This effort is beyond the scope of this study.

13. Did the authors test the rv0455c mutant derived from virulent Mtb in other stressful conditions other than iron limitation?

The reviewer is correct that Rv0455c may have functions in addition to its role in siderophore secretion. This hypothesis is supported by the fact that the siderophore-deficient species *M. leprae* and *M. haemophilum* possess Rv0455c homologs (ML2380 and B586_19750). This possibility is mentioned in the discussion (lines 382-385). Since this is a different project, we did not test other conditions in our study.

Reviewer 2

Overall, the manuscript is well-written, the data clearly presented and interpreted. These results represent a significant advance to the field. The authors have placed their current findings in the context of their previous studies on MmpL4/5 systems. A remaining question is whether Rv0455c interacts with MmpS4/5 or periplasmic domains of MmpL4/5. These experiments are legitimately a lot of work, but if such studies have been attempted it may warrant a mention in the discussion.

We thank the reviewer for the constructive suggestions and comments of our manuscript. We have addressed the suggestions and critiques as follows.

We agree with the reviewer that it would be interesting to examine putative protein interactions of Rv0455c with MmpS4/S5 and MmpL4/L5. However, we did not do these experiments because putative interactions *in vitro* would need to be examined for their relevance in Mtb. Considering that our manuscript already “contains a wealth of information” (reviewer 3), we feel that such an effort would be more appropriate for a follow-up study to identify the molecular function of Rv0455c.

I had a couple of minor suggestions to help the reader follow the story:

Supplemental figure 1, can you add the S15 here to show conservation and genomic context in a broader fashion.

We agree and show the previous figure S15 now as figure S2 in the new version.

I think the authors should decide between Rv0455cSP or Rv0455c31-148 and use consistently.

We use Rv0455cΔSP to emphasize that this Rv0455c protein has no signal peptide (Supplementary Figure S6a). In the protein purification section, we prefer to use Rv0455c₃₁₋₁₄₈ and MSMEG_3494₃₃₋₁₅₃ to define exactly the residues in each protein (lines 220, 230). However, we indicated throughout the manuscript that Rv0455c₃₁₋₁₄₈ and Rv0455cΔSP are the same proteins to avoid confusion.

Reviewer 3

The biochemical and structural aspects of the paper are done very well. The data collection statistics and model refinement seem to be fine for the resolution.

Overall the paper contains a wealth of information about siderophore secretion in a very important pathogen M. tuberculosis. The *in vivo* experiments solidify the importance of siderophore secretion pathways as targets for developing antimicrobial compounds against Mycobacteria. I think this paper will appeal to a wide audience and I recommend its publication after addressing some of my comments.

We thank the reviewer for the comments and the constructive suggestions. We have addressed the critiques as follows.

1) I have a hard time understanding the secretion versus function in the periplasm aspect of the paper. The authors show that majority of the protein is secreted into the media, yet only functional in the periplasm. The model shown in figure 6 does not include this aspect of its function, even though it's unclear presently. They also show that secretion of the protein into the media is not necessary for its function. Could the protein in media be a result of cell lysis?

The fact that Rv0455c is detectable in the Mtb culture filtrate is not due to the cell lysis. In our experiments, we grew the Mtb at a low cell density ($OD_{600} < 0.6$) to minimize the extent of cell lysis (see

Methods, Lines 550-552). The absence of cell lysis was demonstrated using the cytoplasmic GlpX protein as a control, which is not detectable in the culture filtrate of Mtb in our experiments (Fig. 2b).

Our data convincingly show that membrane-anchored Rv0455c is sufficient to restore growth of the Δ *rv0455c* mutant in medium containing mycobactin (Fig. 2c). In addition, we now determined the relative quantities of Rv0455c in the periplasm versus the culture filtrate. In contrast to the superficial impression from the Western blots due to the concentration of the culture filtrate, quantitative image analysis and accounting for the different concentration factors showed that approximately equal amounts of Rv0455c protein are in the culture filtrate and cell-associated in Mtb mc²6230. This is now described in the text (lines 136-142). Furthermore, the phenomenon that proteins with functions in the periplasm are also secreted into the culture filtrate is not uncommon for Mtb. For example, the antigen 85 proteins are mycolyltransferases with their only known functions in the periplasm^{23,24}, are also found in large quantities in the culture filtrate²⁵. This statement was added to the text (lines 142-146).

2) Have the authors checked whether the homologues in *M. smegmatis* and *M. abscessus* are also secreted into culture media?

We did not examine secretion of the Rv0455c homologs in other mycobacteria, since our focus was to understand the function of Rv0455c in Mtb. In addition, adding purified Rv0455c or Ms3494 proteins to the culture did not restore growth of the *rv0455c* mutant in low iron medium with MBT (Fig. S6d), demonstrating the secreted Rv0455c is not involved in siderophore secretion by Mtb. This result is described in the results section (lines 170-173).

3) Out of curiosity, have the authors tried to see if the protein can interact with the surface of liposomes using NMR? Perhaps it needs to be immobilized onto a lipid surface for association.

We did not test this hypothesis since endogenous Rv0455c does not bind to membranes (Figs 2b, S7).

Minor comments:

Have the authors performed structure similarity searches with the DALI web server? They describe the structure as a novel fold, but it would be good to specify whether there are any distant structural homologues. Are the two proteins in the ASU identical? Perhaps specify rmsd somewhere (either in the methods section or in a figure legend)

There is no protein structurally similar to MSMEG_3494 (PDB ID: 7REF) according to the DALI server.

Page 7, lines 140, what TM helices were used for the fusion construct? I could not find the info anywhere.

The sequence information is now added in Table S3.

Page 10 lines 214: specify that the structure was solved by X-ray crystallography.

Revised as suggested (line 242).

Page 21, line 493: SHELX not SHELLX, please fix

Revised as suggested (line 523).

Other changes:

We replaced the abbreviations MBT and cMBT with mycobactin and carboxymycobactin, respectively, to increase the readability of our manuscript for a wide audience.

Reviewers' Comments:

Reviewer #1:

Remarks to the Author:

The authors addressed some of the comments. The resolution of the structure and the virulence phenotype are important findings. But the characterization of the mutant phenotype is limited and the main claim about the function of the Rv0455 protein in siderophore secretion is not fully supported by the data.

Specifically, there is no evidence that Rv0455 is involved in mycobactin secretion, and the evidence for a role in carboxymycobactin secretion remains weak. Yes, mycobactin extracts can be utilized by Mtb as iron source, and yes, mycobactin is found in the culture filtrate. However, mycobactin in the culture filtrate is probably not free but incorporated into extracellular membrane vesicles. The authors agreed about this in the study they refer to in their rebuttal, (Wells et al) when they say "the presence of mycobactin in the culture supernatant is most likely caused by partitioning of cell surface-associated mycobactin with the medium in the presence of detergents", and most mycobactin is membrane associated. If there is evidence in the literature for translocation of free mycobactin across the membrane into the periplasm, which could be aided by Rv0455, they should refer to it.

In addition, the HPLC shows that mycobactin does not accumulate in the Rv0455c mutant. The authors attribute this phenotype to feedback inhibition of synthesis. However, the synthesis of both type of siderophores is coupled and co-regulated, so one would expect that the feedback inhibition would also decrease carboxymycobactin synthesis. In contrast, they observe increase not just in carboxymycobactin, but also monodeoxycarboxymycobactin. It is clear from these results that 1) the impact of the Rv0455c on siderophore synthesis is opposite for mycobactin and carboxymycobactin arguing against an identical role of Rv0455c and 2) despite that total cell associated siderophores may be close (90%) to the wild type, the decrease in deoxymycobactin and increase in deoxycarboxymycobactin indicates dysregulated siderophore biosynthesis in the mutant.

In sum, the study does not provide direct evidence for the claimed role of Rv0455 in mycobactin secretion, and the single evidence for carboxymycobactin, that is the increase of this siderophore in mutant cells can be due to upregulation of synthesis as demonstrated by the increase in monodeoxycarboxymycobactin. As suggested previously the authors must quantify secreted carboxymycobactin in the Rv0455 mutant and compare it to the wild type to support their conclusion on a role for Rv0455 in siderophore secretion. The data does not support the title and subtitles of the manuscript about siderophore secretion. Rv0455 seems to be involved in exogenous siderophore detoxification and virulence but the evidence for involvement in siderophore secretion is suggestive for carboxymycobactin and not there for mycobactin.

The authors should consider other explanations for the toxicity of exogenous siderophores. The toxicity of siderophores could be due to their unique amphiphilic nature and their interaction with the cell envelope, which could be altered by the Rv0455 mutation. To conclusively support their claim that accumulated, not recycled carboxymycobactin in the Rv0455 mutant is toxic, the authors should test the dependence of this phenotype on carboxymycobactin uptake in a double IrtAB-Rv0455 mutant grown with heme.

Related to the previous point, although, the authors do not comment on this, it is obvious from the data that the mutant has a deficiency in heme utilization compared to the wild type. This supports the possibility that the Rv0455 has a more global role in cell envelope transport or integrity.

In this regard, previously, I asked about the fitness of the mutant, but the authors did not properly address this comment. Lines 382-385 are not related to this point and according to their rebuttal they consider the fitness of the mutant as a separate project. A better characterization of the mutant sensitivity to other stresses particularly cell envelope stress is relevant to fully understand the function of Rv0455, and interpret the results of attenuation in vivo, which are revealing. Without those experiments there is no support for the conclusion given by the authors in the in the final paragraph "we identify siderophore poisoning as the mechanism of the large virulence loss of the Mtb mutant lacking the rv0455c".

I noticed that 10 times more mycobactin is used in the toxicity assay in the Msmeg mutant compared to the Mtb mutant. Is this a reflection of higher tolerance in the Msmg mutant to exogenous mycobactin? If that is the case, do the authors have any thoughts about that? Perhaps differences in the cell envelope between Mtb and Msmeg which is related to the previous point.

The authors addressed the question of the discrepancy between the protein localization of the endogenous protein and the postulated function in the periplasm by indicating that after applying dilution factors and imaging, they have now determined that approximately 50 to 60% of Rv0455 was detected in the total cell lysate. This is the opposite to what the western blot shows. The way they get these numbers should be described in the materials and methods and the data presented in way that reflects those numbers.

Rebuttal – Manuscript NCOMMS-21-31817

Manuscript title: A periplasmic cinched protein is required for siderophore secretion and virulence of *Mycobacterium tuberculosis*

Authors: Zhang et al.

Reviewer 1

The authors addressed some of the comments. The resolution of the structure and the virulence phenotype are important findings. But the characterization of the mutant phenotype is limited and the main claim about the function of the Rv0455 protein in siderophore secretion is not fully supported by the data.

Specifically, there is no evidence that Rv0455 is involved in mycobactin secretion, and the evidence for a role in carboxymycobactin secretion remains weak. Yes, mycobactin extracts can be utilized by Mtb as iron source, and yes, mycobactin is found in the culture filtrate. However, mycobactin in the culture filtrate is probably not free but incorporated into extracellular membrane vesicles. The authors agreed about this in the study they refer to in their rebuttal, (Wells et al) when they say “the presence of mycobactin in the culture supernatant is most likely caused by partitioning of cell surface-associated mycobactin with the medium in the presence of detergents”, and most mycobactin is membrane associated.

We now directly determined the secretion of ¹⁴C-labeled siderophores into the culture filtrate of *M. tuberculosis* by thin-layer chromatography. These new experiments showed a **drastic reduction (by >90%) of both carboxymycobactin and mycobactin in the rv0455c deletion mutant of M. tuberculosis.** This experiment is described in **lines 181-205** and shown in the **new Figure 3.**

It is unclear how much mycobactin is secreted in vesicles, in mycobactin micelles, bound to proteins or in detergent micelles. However, even in the absence of detergents plenty of mycobactin is present in the supernatant of Mtb (*PNAS* 113:E348-57;2016), demonstrating that a large fraction of mycobactin is truly secreted. In any of these cases mycobactin must have been transported across the outer membrane and, therefore, is extracellular. This is the very definition of secretion. Secreted molecules can either be released into the extracellular medium or can be attached to the cell surface. A prominent example are the bacterial type V protein secretion systems (autotransporters). In four out of the five subtypes the secreted passenger domain (often toxins) remains attached to the cell surface (*Philos Trans R Soc Lond B Biol Sci* **367**: 1088-1101; 2012).

If there is evidence in the literature for translocation of free mycobactin across the membrane into the periplasm, which could be aided by Rv0455, they should refer to it.

There is no experimental evidence to show how mycobactin is translocated across the inner or outer membranes. This is exactly the mechanism we try to identify. The discovery of Rv0455c is one step closer to this goal.

In addition, the HPLC shows that mycobactin does not accumulate in the Rv0455c mutant. The authors attribute this phenotype to feedback inhibition of synthesis. However, the synthesis of both type of siderophores is coupled and co-regulated, so one would expect that the feedback inhibition would also decrease carboxymycobactin synthesis. In contrast, they observe increase not just in carboxymycobactin, but also monodeoxycarboxymycobactin. It is clear from these results that 1) the impact of the Rv0455c on siderophore synthesis is opposite for mycobactin and carboxymycobactin arguing against an identical role of Rv0455c and 2) despite that total cell associated siderophores may be close (90%) to the wild type, the decrease in deoxymycobactin and increase in deoxycarboxymycobactin indicates dysregulated siderophore biosynthesis in the mutant.

The lipidomics experiments were done with cell-associated siderophores. The reviewer is correct that the quantities of both carboxymycobactins and mycobactins are reduced in the siderophore secretion mutants such as the $\Delta mmpS4/S5$ mutant (*PLoS Pathog* 9: e1003120; 2013). However, the reviewer overlooked that the majority of carboxymycobactins is secreted, while the majority of mycobactins is cell-associated in *M. tuberculosis* (*PLoS Pathog* 9: e1003120; 2013). This means that impaired secretion leads to a relative accumulation of carboxymycobactins despite the overall reduction in siderophore biosynthesis. The same effect is observed for deoxycarboxymycobactins and deoxymycobactins. Identical phenotypes were observed for the siderophore secretion deficient $\Delta mmpS4/S5$ mutant in *M. tuberculosis* (*Proc Natl Acad Sci U S A* 111: 1945-1950; 2014). This is described in the manuscript (lines 232-235). In addition, reduced production of enterobactin siderophores was also observed in enterobactin secretion deficient mutants of *E. coli*. This fact is now described in the discussion (lines 333-335). Thus, **all phenotypes observed in the rv0455c mutant of *M. tuberculosis* are consistent with those observed for other siderophore-deficient mutants of both in *M. tuberculosis* and in *E. coli*.**

In sum, the study does not provide direct evidence for the claimed role of Rv0455 in mycobactin secretion, and the single evidence for carboxymycobactin, that is the increase of this siderophore in mutant cells can be due to upregulation of synthesis as demonstrated by the increase in monodeoxycarboxymycobactin.

As suggested previously the authors must quantify secreted carboxymycobactin in the Rv0455 mutant and compare it to the wild type to support their conclusion on a role for Rv0455 in siderophore secretion. The data does not support the title and subtitles of the manuscript about siderophore secretion. Rv0455 seems to be involved in exogenous siderophore detoxification and virulence but the evidence for involvement in siderophore secretion is suggestive for carboxymycobactin and not there for mycobactin.

The new TLC experiments with ^{14}C -labeled siderophores show strong reduction of both mycobactins and carboxymycobactins in the culture filtrate of *M. tuberculosis*, providing direct evidence for the essential role of Rv0455c in siderophore secretion by *M. tuberculosis* (**new Fig. 3**). This is described in **lines 181-205**.

The authors should consider other explanations for the toxicity of exogenous siderophores. The toxicity of siderophores could be due to their unique amphiphilic nature and their interaction with the cell envelope, which could be altered by the Rv0455 mutation. To conclusively support their claim that accumulated, not recycled carboxymycobactin in the Rv0455 mutant is toxic, the authors should test the dependence of this phenotype on carboxymycobactin uptake in a double IrtAB-Rv0455 mutant grown with heme.

The TLC experiment clearly shows that loss of Rv0455c strongly reduces siderophore secretion (**new Fig. 3**), demonstrating the essential role of Rv0455c in siderophore secretion by *M. tuberculosis*. Thus, additional experiments to explore potential other phenotypes are not warranted.

Related to the previous point, although, the authors do not comment on this, it is obvious from the data that the mutant has a deficiency in heme utilization compared to the wild type. This supports the possibility that the Rv0455 has a more global role in cell envelope transport or integrity.

The reviewer is incorrect. Our previous TnSeq study clearly shows that *rv0455c* gene is dispensable for the growth of Mtb with hemin or hemoglobin (*PLoS Pathog* **16**:e1008337; 2020). The reduced growth of the $\Delta rv0455c$ or the $\Delta mmpS4/S5$ mutants compared to the parent strain is due to the self-poisoning by endogenous siderophores as shown previously (*Proc Natl Acad Sci U S A* **111**: 1945-1950; 2014).

In this regard, previously, I asked about the fitness of the mutant, but the authors did not properly address this comment. Lines 382-385 are not related to this point and according to their rebuttal they consider the fitness of the mutant as a separate project. A better characterization of the mutant sensitivity to other stresses particularly cell envelope stress is relevant to fully understand the function of Rv0455, and interpret the results of attenuation in vivo, which are revealing. Without those experiments there is no support for the conclusion given by the authors in the in the final paragraph “we identify siderophore poisoning as the mechanism of the large virulence loss of the Mtb mutant lacking the *rv0455c*”.

We do not exclude the possibility that the loss of Rv0455 or MmpS4/S5 might have side effects in addition to their essential roles in siderophore secretion by *M. tuberculosis*. However, exploration of other phenotypes and their mechanistical understanding would not be helpful in this study and are, thus, not warranted. There is clearly a direct connection between the *in vitro* toxicity of siderophores, the observed self-poisoning of *M. tuberculosis* by endogenously produced siderophores and the virulence defect of siderophore secretion-deficient mutants of *M. tuberculosis*. To account for the possibility that other, yet unidentified functions of Rv0455c and MmpS4/S5 may contribute to the virulence defect of these *M. tuberculosis* mutants in mice we have rephrased this conclusion to “Furthermore, we identify siderophore poisoning as an important mechanism of the large virulence loss of the *M. tuberculosis* mutant lacking the *rv0455c* gene...” (lines 425-428).

I noticed that 10 times more mycobactin is used in the toxicity assay in the Msmeg mutant compared to the Mtb mutant. Is this a reflection of higher tolerance in the Msmeg mutant to exogenous mycobactin? If that is the case, do the authors have any thoughts about that? Perhaps differences in the cell envelope between Mtb and Msmeg which is related to the previous point.

We did not determine the minimal inhibitory concentration of mycobactin for the *M. smegmatis* *msmeg_3494* mutant and, therefore, do not know whether *M. tuberculosis* is more susceptible to siderophore poisoning than *M. smegmatis*. This is a minor technical point, whose clarification would not contribute anything to this study.

The authors addressed the question of the discrepancy between the protein localization of the endogenous protein and the postulated function in the periplasm by indicating that after applying dilution factors and imaging, they have now determined that approximately 50 to 60% of Rv0455 was detected in the total cell lysate. This is the opposite to what the western blot shows. The way they get these numbers should be described in the materials and methods and the data presented in way that reflects those numbers.

We agree that the Western blots to analyze the subcellular fractions of *M. tuberculosis* are confusing. The main problem is that the band intensities of the cellular fractions cannot be directly compared with those in the culture filtrate, because the culture filtrate needs to be concentrated to visualize the proteins. Quantitative comparison requires the inclusion of the concentration factor. Since the band intensities in the Western blot are misleading, we added all Western blots used for the quantification of the fractions for all examined *M. tuberculosis* strains in Fig. S7. We also added a more detailed description of these experiments to the Method section (lines: 620-643) and added how much sample was loaded for each fraction in the legends of Fig. 2 and Fig. S7.

Other comments and changes:

1. We thank the reviewer for insisting on showing direct evidence for the siderophore secretion defect of the $\Delta rv0455c$ mutant of *M. tuberculosis* instead of relying on the comparison with the $\Delta mmpS4/S5$ mutant. The addition of the analysis of the culture filtrate by TLC experiments providing direct evidence for the essential function of Rv0455c in siderophore secretion by *M. tuberculosis* has improved the manuscript.
2. The entire manuscript was carefully edited again.
3. All changes are marked in a pdf file named "MtbRv0455c_Manuscript_v110_Changes.pdf".

Reviewers' Comments:

Reviewer #1:

Remarks to the Author:

The authors have address my previous comments appropriately.